# Seed coat-derived brassinosteroid signaling regulates endosperm development

Rita B. Lima[1], Rishabh Pankaj [1], Sinah T. Ehlert[1,2], Pascal Finger[1,2], Anja Fröhlich[1], Vincent Bayle [3], Benoit Landrein [3], Arun Sampathkumar [1] & Duarte D. Figueiredo [1] ✉

An angiosperm seed is formed by the embryo and endosperm, which are direct products of fertilization, and by the maternal seed coat. These tissues communicate with each other to ensure synchronized seed development. After fertilization, auxin produced in the endosperm is exported to the integuments where it drives seed coat formation. Here, we show that the seed coat signals back to the endosperm to promote its proliferation via the steroid hormones brassinosteroids (BR). We show that BR regulate cell wall-related processes in the seed coat and that the biophysical properties of this maternal organ determine the proliferation rate of the endosperm in a manner independent of the timing of its cellularization. We thus propose that maternal BR signaling tunes endosperm proliferation to seed coat expansion.

In most angiosperms, seed development is coupled to fertilization, where its two products, endosperm and embryo, are surrounded by a tissue of maternal sporophytic origin, the seed coat. The communication between these three structures, embryo, endosperm and seed coat, is imperative for successful seed formation[1,2]. In particular, the endosperm and the seed coat develop in a coordinated manner, which in the early stages of seed development seems to be mostly independent of embryo development[3]. This coordination starts at very early stages of seed development. Before fertilization, the development of both endosperm and seed coat is blocked by the POLYCOMB REPRESSIVE COMPLEX 2 (PRC2)[4]. In the ovule central cell, the FERTILIZATION INDEPENDENT SEED-PRC2 (FIS-PRC2) blocks the expression of maternal alleles of auxin biosynthesis genes[5]. The post-fertilization production of auxin, driven by the paternal alleles of those genes, leads to endosperm formation[5], and auxin is transported to the ovule integuments, allowing for the seed coat to develop[6]. Consequently, loss of PRC2 or exogenous auxin applications induces seed development without fertilization[4–10]. Also importantly, interactions between seed coat and endosperm dictate seed size. For example, increased seed coat size is accompanied by an increase in the size of the endosperm cavity[11]. On the other hand, smaller endosperms, like those produced by *miniseed 3* (*mini3*) and *haiku 2* (*iku2*), lead to reduced seed coat growth[12].

In most flowering plants, endosperm development starts as a coenocyte, which later cellularizes. In *Arabidopsis thaliana* (Arabidopsis), this coincides with the embryo at the early heart stage and with a decrease in seed expansion rates[13,14]. The timing of cellularization was shown to correlate with the endosperm proliferation rate: mutants with reduced endosperm proliferation typically show early cellularization phenotypes[12], while those with proliferative endosperms often display late cellularization[15]. In addition, endosperm proliferation and timing of cellularization also correlate with seed coat size: *APETALA 2* (*ap2*) mutants produce larger seed coats, and have more proliferative endosperms that are associated with delayed cellularization[16]. In contrast, *TRANSPARENT TESTA GLABRA2* (*ttg2*) mutants produce smaller seeds with smaller and less proliferative endosperms, that initiate cellularization earlier than the WT[17]. Thus, the rate of endosperm proliferation, together with the size of the seed coat, was proposed to determine how much the endosperm grows by setting the timing of cellularization. However, although the onset of cellularization coincides with the end of coenocytical endosperm proliferation, inhibition of seed coat growth actually precedes it[18], meaning that these two processes can be unlinked.

Plant organ growth is dependent on two main processes: cell division and cell elongation. One of the endogenous factors

[1]Max Planck Institute of Molecular Plant Physiology, Potsdam Science Park, 14476 Potsdam, Germany. [2]Institute of Biochemistry and Biology, University of Potsdam, 14476 Potsdam, Germany. [3]Laboratoire Reproduction et Développement des Plantes, Univ. Lyon, ENS de Lyon, UCB Lyon 1, CNRS, INRAE, INRIA, 69364 Lyon, France. ✉e-mail: figueiredo@mpimp-golm.mpg.de

contributing to these two processes are Brassinosteroids (BR). For example, exogenous application of low concentrations of BR increases meristem cell proliferation, whereas higher concentrations inhibit it, suggesting a dose-dependent response[19]. The same is observed for pollen tube growth[20]. Given that pollen tubes are single cells, BR function in this process consists of exclusively promoting cell elongation[20]. Consistently, BR signaling has been linked to cytoskeleton dynamics and cell wall remodeling to control cell elongation[21,22]. BR-mediated changes to microtubule cytoskeleton likely result in changes to the deposition of cellulose microfibrils, which in turn impacts organ growth and the mechanical forces existing within cells[23]. Importantly, mechanical forces play a role in the communication between seed coat and endosperm. The cell wall, which separates the outer and the inner integuments of the seed coat, was reported to have mechanosensitive properties, and its mechanical properties are non-cell autonomously regulated by endosperm expansion[24]. The rapid growing phase of the coenocytic endosperm associates with high levels of turgor pressure that progressively decrease throughout seed development[13]. This decrease in pressure was initially thought to lead to slower seed expansion[13]. However, turgor pressure from the endosperm, although initially promoting seed coat growth, also indirectly restricts it by inducing stiffening of the walls of the mechanosensitive layer of the seed coat[18]. Therefore, a reduction in endosperm turgor pressure avoids the precocious stiffening of the seed coat. Turgor driven growth in itself could influence the integrity of the cell wall, requiring the presence of surveillance machinery to ensure proper cell wall homeostasis during growth. Moreover, changes to the integrity and composition of the cell wall are known to trigger BR signaling[25], indicating the existence of a complex feedback mechanism between mechanical forces, growth and hormone signaling.

Recent studies thus show that endosperm pressure can affect seed coat development through mechanical forces. However, to which extent seed coat properties can, in turn, affect endosperm growth and development has not been studied. Here, we tested BR as a seed coat-derived signal that regulates endosperm formation. We demonstrate that the physical size of the seed coat determines endosperm proliferation rates and in a manner independent from the timing of its cellularization. We thus propose that the biophysical properties of the seed coat regulate endosperm proliferation and that maternal BR signaling is necessary for the coordination of endosperm proliferation to seed coat growth.

## Results

### BR are required for endosperm development

We hypothesized that a seed coat-derived signal would be involved in promoting endosperm proliferation. This hypothesis was based on observations made in PRC2 mutants in Arabidopsis (Fig. 1a–f). Mutants specifically lacking gametophytic PRC2 function, like *fertilization independent seed 2* (*fis2*), produce asexual (or autonomous) endosperms at a low frequency, and those endosperms do not proliferate much (Fig. 1b). If it is true that seed coat formation promotes endosperm proliferation, then inducing seed coat formation in a FIS-PRC2 mutant should enhance asexual endosperm formation. The mutant *fertilization independent endosperm* (*fie*) is depleted in both gametophytic and sporophytic PRC2s, and thus both the endosperm and the seed coat develop autonomously (unlike *fis2*, where only the endosperm develops)[4]. Consistent with our hypothesis, *fie* mutants produce more autonomous endosperms than *fis2*, and those endosperms are more proliferative (Fig. 1c). Thus, endosperm formation due to loss of FIS-PRC2 is enhanced by the presence of a developing seed coat[4]. We previously showed that exogenous auxin is also sufficient to initiate fertilization-independent endosperm formation[5]. However, those autonomous endosperms, like those of *fis2*, are small and do not proliferate like those of *fie* (Fig. 1d). A quantification of

endosperm nuclei number and of autonomous seed size can be seen in Fig. 1e, f, demonstrating that indeed loss of sporophytic PRC2 in *fie* mutants results in more proliferative endosperms and in bigger autonomous seeds, when compared to *fis2* or with auxin-derived autonomous seeds. This indicates that seed coat-derived signals promote and sustain endosperm growth. Because there are no cytoplasmic connections between endosperm and seed coat, these tissues can only communicate via signals that can cross membranes, like hormones or peptides[26]. We analyzed public datasets and found that genes involved in BR biosynthesis and signaling are predicted to be expressed in the endosperm and in the seed coat (Supplementary Fig. 1)[27]. We thus hypothesized that BR could be a seed coat-derived signal that promotes endosperm proliferation. To determine if BR have a role during endosperm development, we investigated whether endosperm proliferation was affected in mutants affected in BR biosynthesis or signaling. We screened several loss-of-function BR mutants to assess their seed set and discovered that several produced malformed ovules (Supplementary Fig. 1). This severely limits their seed set and is therefore not useful for our purposes. Thus, throughout this work, we mostly focused on the *deetiolated2-1* (*det2-1*) and *bassinosteroid insensitive1-6* (*bri1-6*) mutants, which, although impaired in BR biosynthesis and signaling, respectively, are fully fertile[28,29]. Figure 1g, h shows simplified BR biosynthesis and signaling pathways, to indicate which steps are affected in the mutants analyzed here.

If BR are required for endosperm development, mutations in genes encoding enzymes required for BR biosynthesis are expected to affect its proliferation. We thus scored the number of endosperm nuclei produced in the BR biosynthesis mutant *det2-1* at 2 days after pollination (2 DAP). We chose this time point as it allows for a large-scale quantification of endosperm nuclei in different genotypes. For all assays, we only assessed seeds whose embryos were at the 2-celled stage, to rule out variation due to differences in pollination efficiency. As hypothesized, *det2-1*$^{-/-}$ seeds produce fewer endosperm nuclei compared to the wild-type (WT) (Fig. 1i–k). To validate this, we tested additional BR biosynthesis mutants. We obtained the double mutant *br6ox1*$^{-/-}$ *br6ox2*$^{-/-}$, which is impaired in the two last steps of BR biosynthesis[30–32], and again observed reduced endosperm proliferation, in comparison to the WT (Fig. 1i). Similar results were obtained for the BR biosynthesis mutant *cpd*[33] (Fig. 1i). We then tested if increased BR levels would yield the opposite phenotype, i.e., increased endosperm proliferation. For this, we used the BR-overproducing mutant *dwf4-5D*[34]. However, we saw no differences between this mutant and the WT (Fig. 1i), which could point to the saturation of the BR receptors in the WT condition.

We then tested if similar phenotypes were observed for mutants impaired in BR signaling. Indeed, mutants lacking the main BR receptor BRI1 (*bri1-6*$^{-/-}$, Fig. 1h) also revealed less endosperm nuclei than the WT (Fig. 1i). Next, we investigated endosperm development in lines mutant for genes encoding BR signaling effector transcription factors (TFs; Fig. 1h). Nevertheless, we did not observe endosperm defects in single mutants of the main BR effectors, *bri1-ems-suppressor1-2* (*bes1-2*$^{-/-}$) or *brassinazole-resistant1-2*$^{-/-}$ (*bzr1-2*; Supplementary Fig. 2). The same was true for *bzr1-homologue2* (*beh2-1*$^{-/-}$; Fig. 1j), in which the *BEH* with the strongest expression in seeds is knocked-out (Supplementary Fig. 2). To account for possible functional redundancy between these TFs, we analyzed *bes1-p*, which is a pentuple mutant for several members of the BZR/BES family (*bzr1bes1beh1beh3beh4*)[35]. Again, we did not observe significant differences compared to the WT (Fig. 1j). However, the expression of *BZR1* and *BES1* is only slightly reduced in this mutant, and *BEH2* is still present[35]. This may explain why we did not observe any obvious phenotypes even in this higher-order mutant. Overall, our data indicates that signaling via BRI1 is necessary for proper endosperm development, but it is unclear which of the known BR effector TFs are involved in the process.

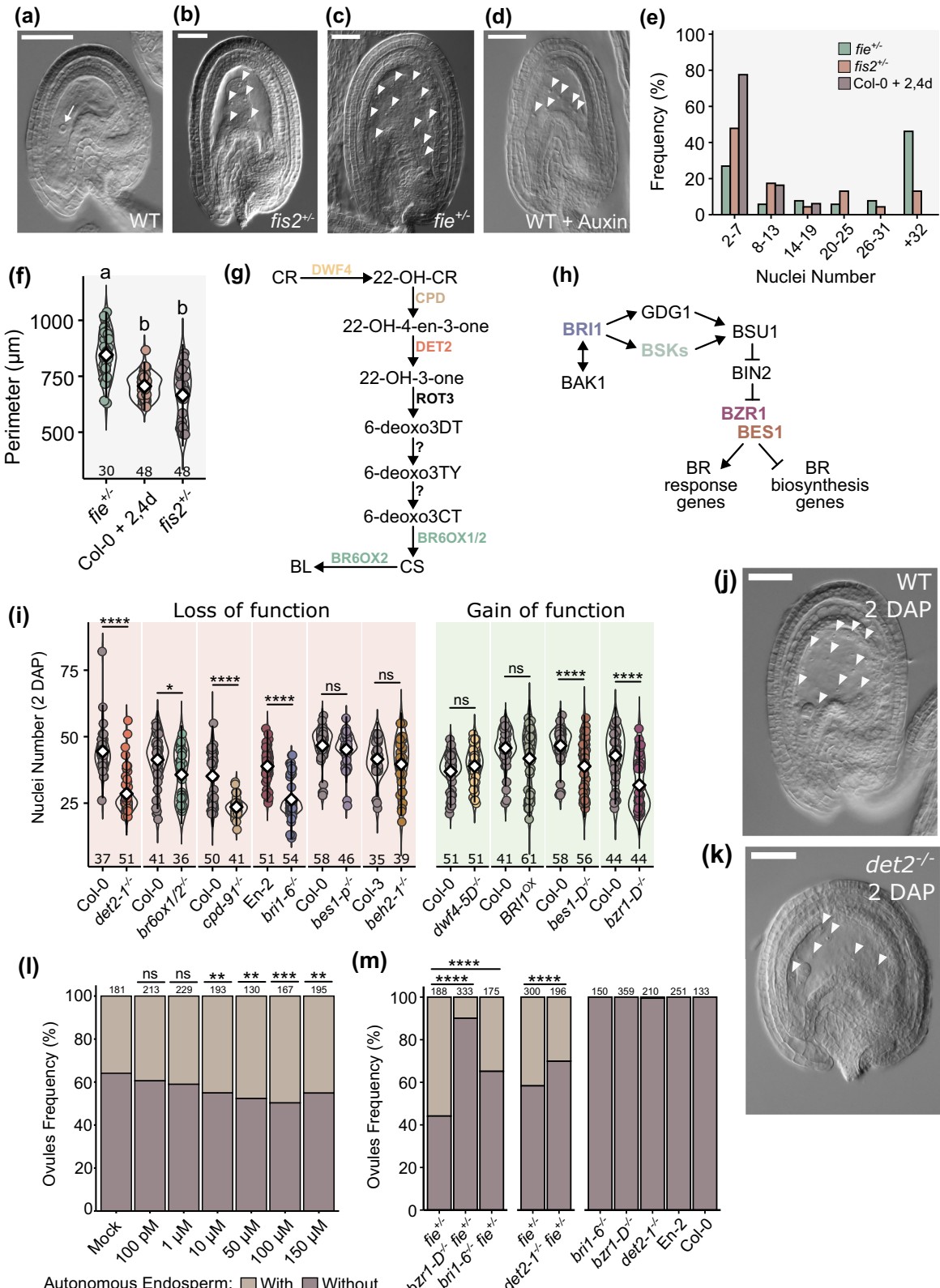

We then hypothesized that mutations resulting in constitutive BR signaling could lead to increased endosperm proliferation. Therefore, we analyzed lines carrying gain-of-function alleles in BR effectors: *BRI1^OX*, *bes1-D* and *bzr1-D* [36–39]. While we observed no differences between *BRI1^OX* and WT, *bes1-D^−/−* and *bzr1-D^−/−* unexpectedly showed a significant reduction in endosperm proliferation (Fig. 1i). We, however, noted that these mutants also exhibited malformed

ovules (Supplementary Fig. 2). Given that BZR1 and BES1 down-regulate BR biosynthesis genes[40], we questioned if these phenotypes could be due to decreased endogenous BR. To test this, we generated the double mutant *bzr1-D^−/−dwf4-5D^−/−*, as *DWF4* is a target of BZR1 repression[40,41]. However, we saw no differences between *bzr1-D^−/−dwf4-5D^−/−* and the single mutant *bzr1-D^−/−* (Supplementary Fig. 2), indicating that the decrease in nuclei number observed in *bzr1-D^−/−* is

**Fig. 1 | BR are necessary for endosperm development. a–d** Microscopic photos of WT unfertilized ovule (**a**), PRC2 mutant seeds (**b**, **c**), and an auxin-induced autonomous seed (**d**). Scale bars, 50 μm. This experiment was repeated three times with similar results. **e**, **f** Quantification of endosperm nuclei and size in seeds of the genotypes indicated in panels (**b**–**d**). Error bars represent standard deviation. The letters on top indicate significance of differences by one-way analysis of variance (ANOVA). The number below each sample indicate number of seeds analyzed. **g**, **h** Simplified BR biosynthesis (**g**) and signaling (**h**) pathways. **i** Endosperm nuclei number 2 days after pollination (2 DAP) for BR mutants and respective WT. Error bars represent standard deviation. The letters on top indicate significance of

differences by one-way ANOVA. The number below each sample indicate number of seeds analyzed. **j**, **k** 2 DAP seeds of WT (**b**) and *det2-1*$^{-/-}$ (**c**). Scale bars, 50 μm. These experiments were repeated a minimum of three times with similar results. **l** Percentage of ovules developing autonomous endosperm after the application of exogenous auxin (mock) or auxin in combination with different concentrations of epi-brassinolide. **m** Autonomous endosperm development in BR and PRC2 double mutants, and respective single mutants, at 5 DAE. The values on top indicate the number of ovules analyzed. Significance of difference was determined by one-way chi-squared test. ****$p < 0.0001$, ***$p < 0.001$, **$p < 0.01$ and *$p < 0.05$.

likely caused by excessive signaling rather than low endogenous BR. These observations also reveal that constitutive BR signaling is detrimental for ovule development and for endosperm proliferation.

To further corroborate the positive role of BR in endosperm development, we performed exogenous BR applications to develop seeds. We previously showed that exogenous auxin is sufficient to induce autonomous endosperm development[5]. We thus hypothesized that supplying BR in addition to auxin should increase the frequency of ovules producing autonomous endosperm comparatively to when only auxin is provided. Indeed, we observed a gradual and significant increase in the frequency of ovules developing autonomous endosperm when 10 μM to 100 μM of epi-Brassinoline (epi-BR) were applied (Fig. 1l). Finally, we asked if BR mutations also affect the development of autonomous endosperm of PRC2 mutants. We generated *fie det2, fie bri1-6* and *fie bzr1-D* and quantified autonomous endosperm formation in ovules at 5 DAE. Indeed, loss of BR biosynthesis or signaling also led to reduced autonomous endosperm formation in *fie* (Fig. 1m). The same was observed for *bzr1-1D*$^{-/-}$ *fie*$^{+/-}$ (Fig. 1m).

Taken together, our results indicate that BR signaling via BRI1 positively regulates endosperm proliferation and argue for a dose-dependent mode of action in which excessive BR signaling has a negative effect on ovule and endosperm development.

## The timing of endosperm cellularization is not affected in BR mutants

It is thought that both the rate of endosperm proliferation as well as the size of the seed coat determine how much the endosperm grows by fixing the timing of cellularization. Considering that the BR mutants *det2-1* and *bri1-6* exhibit reduced endosperm proliferation, we predicted that endosperm cellularization would also initiate earlier in these mutants. Surprisingly, we did not observe any difference in the timing of cellularization between them and the respective WTs (Fig. 2). Just like the WT seeds, both *det2-1*$^{-/-}$ and *bri1-6*$^{-/-}$ endosperms initiated cellularization when the embryo was at the early heart stage (Fig. 2). Similarly, by the time the embryos were in the torpedo stage, almost the whole seed cavity had undergone cellularization. These observations indicate that the timing of endosperm cellularization can be uncoupled from the rate of endosperm proliferation.

## BR contribute to endosperm development via the maternal sporophytic tissues

Genome-wide profiling of gene activity in Arabidopsis seeds suggested that genes involved in BR perception and biosynthesis are expressed in the endosperm and seed coat (Supplementary Fig. 1)[27]. To validate this, we analyzed published reporter lines[20,42,43], and developed transcriptional reporters for the genes encoding the receptors BRI1 and BRI1-ASSOCIATED RECEPTOR KINASE1 (BAK1) and for the biosynthetic gene *DET2*. All reporters were imaged in unfertilized ovules and in seeds at 1 DAP (Fig. 3a and Supplementary Fig. 3), as we expected these genes to be expressed at time points preceding the phenotypes described in Fig. 1. Indeed, we observed expression of all enzymes that participate in BR biosynthesis in the sporophytic tissues in both time points. The biosynthetic pathway

can be seen in Fig. 1g. Interestingly, these enzymes were expressed in different domains of the seed coat: DWF4-GFP was confined to the outer integument layers and to the endothelium; the *CPD* transcript was detected in the micropylar region; *DET2* was expressed exclusively in the outer layers of the integuments and seed coat; ROT3-GFP was only expressed in the funiculus; and BR6OX1-GFP and BR6OX2-GFP were both expressed in the chalazal region, although BR6ox1-GFP was also detected close to the micropyle and BR6OX2-GFP in the outer integument (Fig. 3a and Supplementary Fig. S3). Additionally, the BR6OX2 reporter was occasionally detected in the synergids (Fig. 3a), and rarely in the central cell and in the endosperm (Supplementary Fig. 3), making it the only reporter construct that we detected in the gametophyte or zygotic products. The most striking observation was that most genes necessary for BR biosynthesis were expressed in the sporophytic tissues of the ovule, suggesting that the integuments and seed coat are likely the main source of BR during early seed formation. However, these observations raise the question of how BR intermediates are transported between different layers of the seed coat. Recently, the ABCB-type transporters ABCB1 and ABCB19 (also known as PGP1 and PGP19) were shown to export BRs[44]. Fitting with the expression of BR biosynthesis machinery, we found both exporters expressed in several layers of the seed coat, but never in the endosperm (Supplementary Fig. 3). ABCB1 is weakly expressed in the inner layers of the seed coat, and only post-fertilization, while ABCB19 is already expressed in the ovule integuments and its expression pattern remains comparatively strong after fertilization (Supplementary Fig. 3). Thus, we propose that BRs are produced in the seed coat during these early stages of seed development and that ABCB1/19 are involved in the transport of BR intermediates between the different layers of the seed coat.

As for the components of the BR signaling pathway (Fig. 1h), *BRI1*, *BAK1*, *BZR1* and *BES1* were all expressed in the integuments before fertilization and in the seed coat at 1 DAP (Fig. 3a and Supplementary Fig. 3). *BRI1*, *BAK1* and *BZR1* were expressed in all cell layers of the integuments and of the seed coat, while BES1-GFP was mostly observed in the chalaza (Fig. 3a). Importantly, no expression was detected in the central cell or endosperm for any of the reporters, suggesting that BRs are produced and perceived in the sporophytic tissues of the ovule/seed, and regulate endosperm development indirectly.

To test if the effect of BRs in endosperm proliferation is sporophytic, we performed reciprocal crosses between WT and BR mutants. Indeed, we observed that *bri1-6*$^{-/-}$ seeds fertilized by WT pollen show a significant decrease in the number of endosperm nuclei, comparatively to selfed WT. When *bri1-6*$^{-/-}$ was used as a father, however, the resulting seeds were not different from the WT (Fig. 3b). These observations agree with a sporophytic regulation of endosperm development by BR signaling via BRI1. However, for the biosynthesis mutant *det2-1*$^{-/-}$, we observed a reduction in endosperm nuclei number when the mutant was used either as a mother or a father (Fig. 3c). The phenotype was exacerbated in selfed *det2-1*$^{-/-}$ seeds, indicating that *det2-1*$^{-/-}$ pollen further enhanced the defects in endosperm proliferation. This suggests a possible gametophytic or zygotic effect of DET2 in endosperm development. Thus, to challenge our hypothesis of a

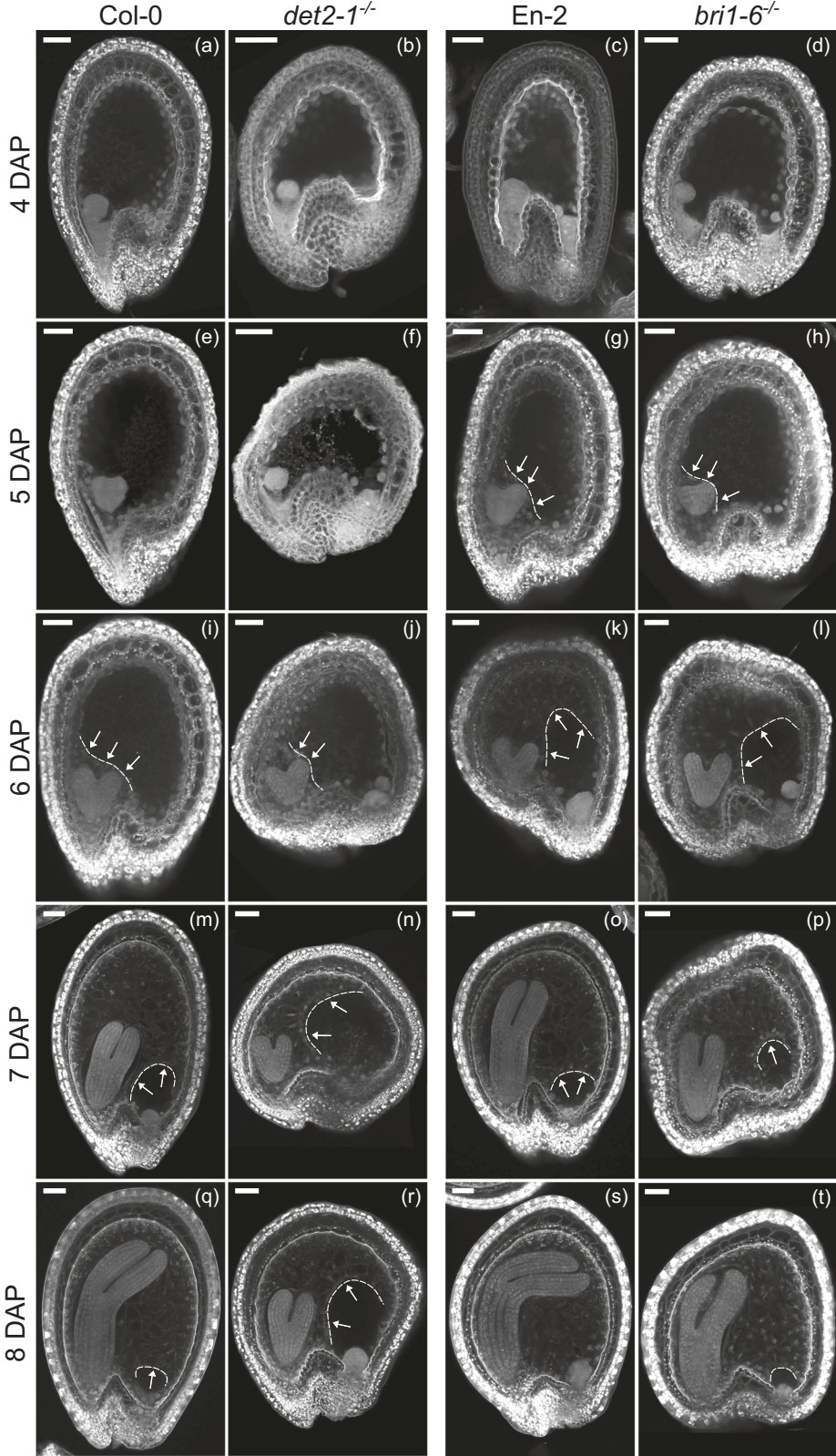

**Fig. 2 | Endosperm cellularization is not affected in BR mutants. a–d** Seeds of Col-0 (**a**), *det2-1⁻ᐟ⁻* (**b**), En-2 (**c**) and *bri1-6⁻ᐟ⁻* (**d**) at 4 DAP with no visible cellularization; **e–h** Seeds of Col-0 (**e**), *det2-1⁻ᐟ⁻* (**f**), En-2 (**g**) and *bri1-6⁻ᐟ⁻* (**h**) at 5 DAP with visible cellular endosperm; **i–l** Seeds of Col-0 (**i**), *det2-1⁻ᐟ⁻* (**j**), En-2 (**k**) and *bri1-6⁻ᐟ⁻* (**l**) at 6 DAP with cellular endosperm around the embryo or further developed; **m–p** Seeds of Col-0 (**m**), *det2-1⁻ᐟ⁻* (**n**), En-2 (**o**) and *bri1-6⁻ᐟ⁻* (**p**) at 7 DAP with cellular endosperm; **q–t** Seeds of Col-0 (**q**), *det2-1⁻ᐟ⁻* (**r**), En-2 (**s**) and *bri1-6⁻ᐟ⁻* (**t**) at 8 DAP with endosperm cellularization ongoing or almost completed. Arrows point to cellular endosperm. Dashed lines delimit endosperm cellularization. Scale bars, 50 µm. These experiments were repeated twice with similar results.

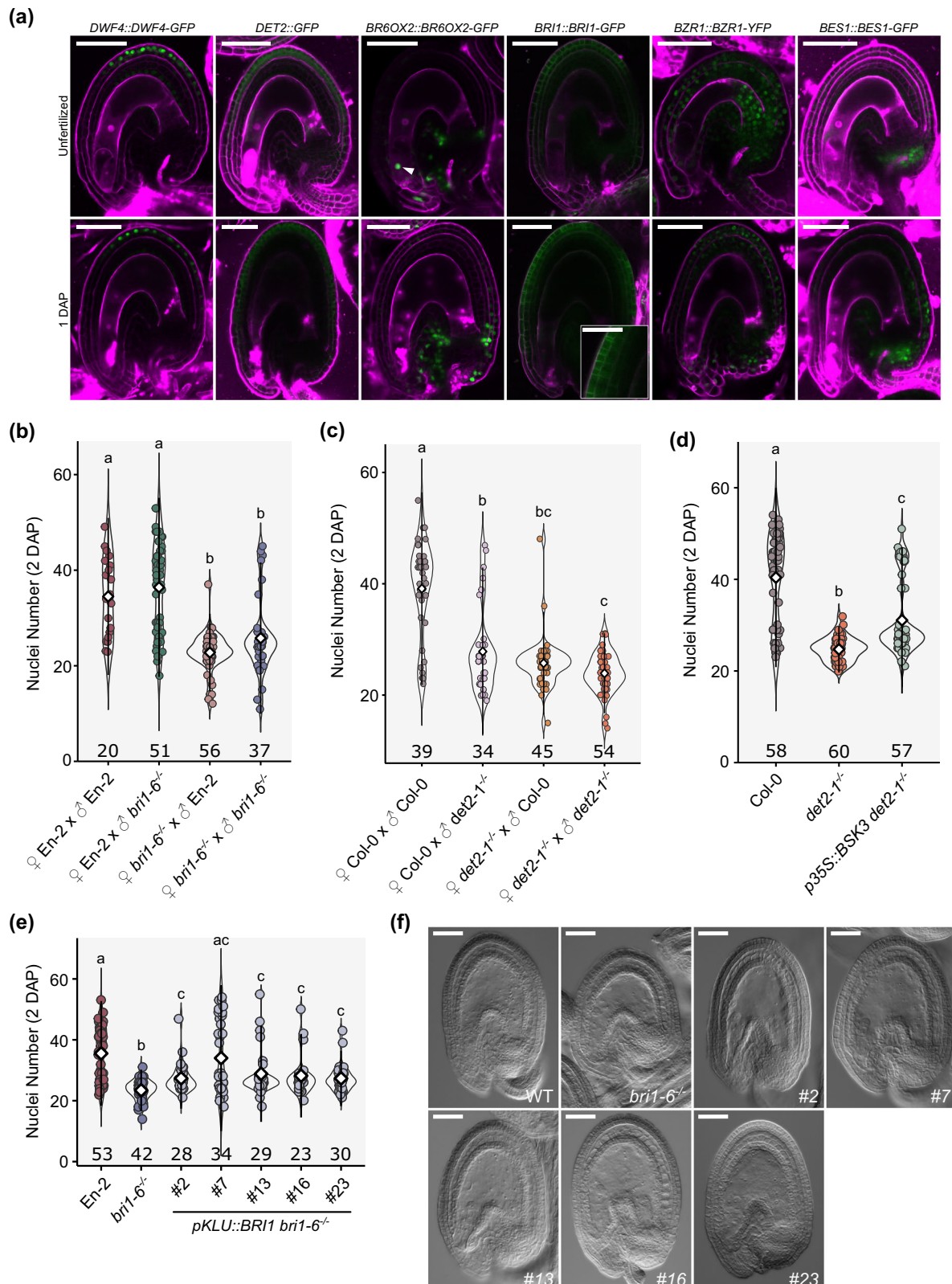

sporophytic action of BR, we performed auxin treatments in $det2\text{-}1^{+/-}$ and $bri1\text{-}6^{+/-}$ ovules. If BR are required in the gametophyte to stimulate endosperm growth, the frequency of autonomous endosperm formation should be significantly reduced in the heterozygotes compared to the WT. In agreement with a sporophytic effect for BR signaling, the frequency of $bri1\text{-}6^{+/-}$ ovules developing autonomous endosperm was similar to the WT (Supplementary Fig. 3). In contrast, $det2\text{-}1^{+/-}$

produced significantly less autonomous seeds compared to the WT (Supplementary Fig. 3), indicating a possible gametophytic effect. These observations argue that while BR signaling via BRI1 affects endosperm development from the sporophytic tissues, BR biosynthesis in the gametophyte may also impact endosperm development. However, we also cannot rule out that the $det2\text{-}1^{+/-}$ mutation is haploinsufficient.

**Fig. 3 | BR effect on endosperm development is sporophytic. a** Expression pattern of *DWF4::DWF4-GFP, DET2::GFP, BR6OX2::BR6OX2-GFP, BRI1::BRI1-GFP, BZR1::BZR1-YFP* and *BES1::BES1-GFP* in unfertilized ovules (upper row) and seeds at 1 DAP (bottom row). Ovules and seeds were stained with propidium iodide (in magenta). Scale bars, 50 μm (25 μm for the inset of BRI1-GFP). These experiments were twice with similar results. **b, c** Quantification of endosperm nuclei at 2 DAP for reciprocal crosses between WT and *bri1-6⁻/⁻* (**b**) and between WT and *det2-1⁻/⁻* (**c**). Error bars represent standard deviation. Significance of differences was determined by one-way ANOVA, followed by Tukey's HSD test (letters). **d** Number of endosperm nuclei in WT, *det2-1⁻/⁻* and *p35S::BSK3 det2-1⁻/⁻* at 2 DAP. Error bars represent standard deviation. Significance of differences was determined by one-way ANOVA, followed by Tukey's HSD test (letters). **e** Number of endosperm nuclei in WT, *bri1-6⁻/⁻* and in five complementation lines expressing *KLU::BRI1* at 2 DAP. Error bars represent standard deviation. The letters indicate statistically significant differences as determined by pairwise one-way Wilcoxon test. The number below each sample indicate number of seeds analyzed. Representative photos of 2 DAP seeds of all genotypes can be seen in (**f**). These experiments were repeated twice with similar results. Scale bars, 50 μm.

To further test if the effect of BR in endosperm development indeed originates in the sporophytic tissues, we postulated that ectopic BR activity in a *det2-1⁻/⁻* mutant background should complement its endosperm defects. Thus, we quantified endosperm formation in *det2-1⁻/⁻* complemented with *CaMV35S::BSK3*[45], as the *CaMV35S* promoter is only active in the sporophytic tissues of the plant. In this line, although BR biosynthesis is compromised, BR signaling is ectopically active in the seed coat, but not in the endosperm. Indeed, *35S::BSK3 det2-1⁻/⁻* seeds revealed a partial but significant increase in endosperm proliferation compared to the single mutant *det2-1⁻/⁻* (Fig. 3d).

To validate the sporophytic effect of BR signaling via BRI1, we complemented the *bri1-6⁻/⁻* mutant with the *BRI1* coding sequence driven by the *KLUH* promoter, which is specifically expressed in the sporophyte[46]. Consistent with a sporophytic role of BR signaling in regulating endosperm development, several independent transgenic lines showed a partial but significant rescue of the *bri1-6⁻/⁻* endosperm phenotypes (Fig. 3e, f). This supports our claim that BR function is necessary for the seed coat to sustain endosperm proliferation.

In conclusion, these results demonstrate that BR produced and perceived in the seed coat contributes to endosperm development. However, we cannot fully exclude that BR biosynthesis and signaling originating in zygotic tissues may also participate in this process.

## BR mutant seeds are affected in cell wall and auxin-related processes

Our data points to BRs acting at least partly via a sporophytic effect in endosperm formation. However, the question remains on what the nature of that signaling mechanism is. To identify the genes that are downstream of BR and that could be involved in regulating endosperm development, we carried out a transcriptomics approach. Our experimental setup consisted of treating WT and *det2-1⁻/⁻* ovules with either auxin or a combination of auxin and epi-BR. Thus inducing autonomous seed development, as done for Fig. 1l. We reasoned that the expression of genes deregulated in *det2-1⁻/⁻* autonomous seeds should be normalized in *det2-1⁻/⁻* mutants exogenously treated with epi-BR. We used autonomous seeds rather than sexual ones to avoid the presence of embryos, which could complicate our analysis. Emasculated pistils were treated with 100 μM of 2,4-D or with 100 μM of both 2,4-D and epi-BR. As expected, we observed a significant increase in the number of ovules producing autonomous endosperm in samples treated with auxin and BR, comparatively to those treated exclusively with auxin (Fig. 4a). In particular, *det2-1⁻/⁻* treated with auxin and epi-BL was indistinguishable from the WT, as we had postulated (Fig. 4a). We thus tested which genes were differentially expressed (DEGs) between *det2-1⁻/⁻* and WT auxin-treated ovules and detected 980 genes downregulated in *det2-1⁻/⁻* (Supplementary Fig. 4; Supplementary Data 1). We also found that 1037 genes were upregulated in *det2-1⁻/⁻* treated with auxin versus auxin and BR (Supplementary Fig. 4; Supplementary Data 1). Moreover, we detected 1037 genes upregulated in *det2-1⁻/⁻* and 1125 genes for WT when each genotype was treated with epi-BL. From these, 412 genes were shared between the datasets (Fig. 4b). We discovered that processes such as cell wall biogenesis, response to auxin, hormone level regulation, and polysaccharide metabolism were significantly enriched

in both genotypes upon epi-BR treatments (Fig. 4c). Among the genes involved in response to auxin we found genes encoding auxin transporters (*PIN-FORMED 1* [*PIN1*], *PIN3, PIN7, LIKE-AUX1 3* [*LAX3*] and *ATP-BINDING SITE CASSETE 1* [*ABCB1*]), biosynthetic enzymes (*YUCCA6* [*YUC6*]) and genes responsive to auxin (*INDOLE-3-ACETIC ACID INDUCIBLE 1* [*IAA1*], *IAA9, IAA11, IAA20* and *IAA31*) (Fig. 4d). As for genes related to cell wall processes, we found proteins involved in cell wall loosening (*EXPANSIN1* [*EXPA1*], *EXPA4, EXPA15*), pectin demethylesterification (*PECTIN METHYLESTERASE 3* [*PME3*], *PME12* and *PME34*) and cellulose biosynthesis (*CELLULOSE SYNTHASE-LIKE A2* [*CSLA02*], *CSLA10* and *COMPANION OF CELLULOSE SYNTHASE 2* [*CC2*]) (Fig. 4d).

In summary, our transcriptomic analysis suggests that BRs likely regulate endosperm development through the modulation of auxin activity and of cell wall properties.

## Auxin signaling is downregulated in the endosperm of BR mutants

Given our transcriptomic analysis and knowing that auxin activity is required for endosperm proliferation[5], we tested if impaired BR function could deregulate auxin activity in the endosperm. For this, we made use of the auxin degradation-based sensor *R2D2*[47]. In the WT, we observed a strong reduction of the VENUS/tdTomato signal ratio in the endosperm, when compared to that of the central cell, indicating increased auxin activity post-fertilization (Fig. 5a, b and Supplementary Fig. 5). However, this sharp decline in the VENUS/tdTomato ratio was not observed in the endosperms of *det2-1⁻/⁻* or of *bri1-6⁻/⁻* (Fig. 5b and Supplementary Fig. 5), indicating that the endosperms of these mutants have reduced auxin activity comparatively to the WT. Thus, it is likely that *det2-1⁻/⁻* and *bri1-6⁻/⁻* seeds produce less endosperm because of reduced auxin activity.

We then tested if auxin activity was also affected in the seed coat of BR mutants. The reasoning for this was that BR mutants have been shown to produce smaller seeds compared to the WT[48,49], which we confirmed (Supplementary Fig. 6). Given that cell expansion can be in part due to the presence of auxin in the apoplast, we questioned if auxin activity was also reduced in the seed coats of *det2-1⁻/⁻* and *bri1-6⁻/⁻* seeds. This reduction would in turn result in auxin-mediated cell expansion and, consequently, smaller seeds with smaller endosperms. However, we did not detect significant differences in R2D2 ratios for *det2-1⁻/⁻*, *bri1-6⁻/⁻* and WT seed coats (Fig. 5b; Supplementary Fig. 5). Thus, BRs affect endosperm proliferation in a manner independently of sporophytic auxin signaling.

## Lack of BR leads to changes in pectin methyl esterification in the seed coat

Our transcriptomic analysis also points to BR regulating cell wall formation and organization. To validate this, we performed cell wall immunostaining to investigate if the components of seed coat cell walls are altered in BR mutants compared to the WT. We used calcofluor white to visualize the cellular outlines and the antibodies LM19 and LM20 to visualize de-methylesterified and methylesterified pectins, respectively. While de-methylesterified pectins contribute to stiffening the cell walls, methylesterified pectins are associated with

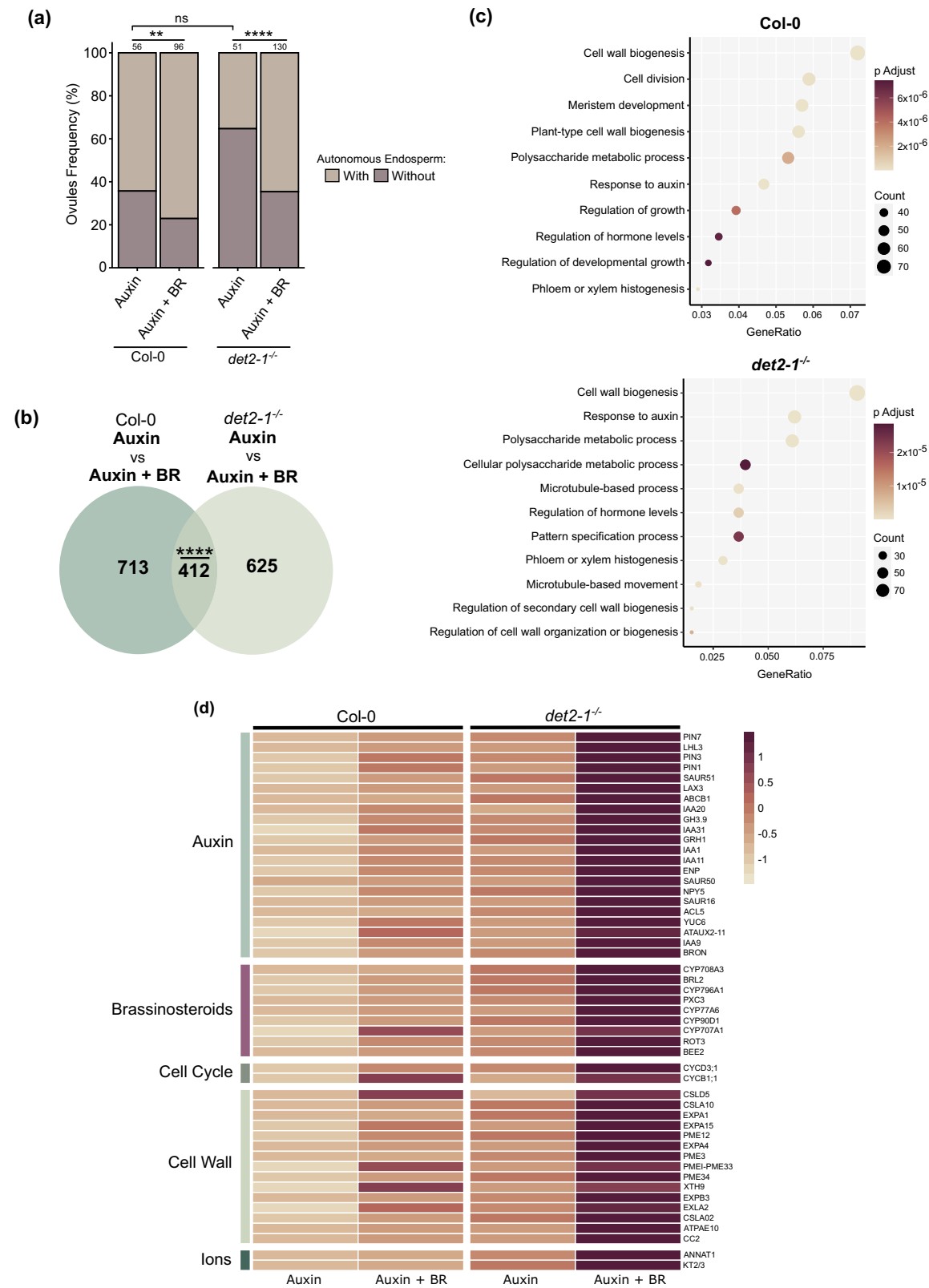

**Fig. 4 | BR regulate hormonal and cell wall-related processes in seeds.**
**a** Percentage of ovules developing autonomous endosperm in WT and *det2-1⁻/⁻*, 3 days after treatment (DAT) with auxin or auxin plus epi-BR. The values on top of each bar indicate the number of ovules analyzed. Significance of differences was determined by one-way chi-squared test. **b** Venn diagram showing overlapping genes that are upregulated in WT (above) and *det2-1⁻/⁻* (below) ovules treated with auxin versus auxin plus epi-BR. Overlap is significant as determined by a one-way hypergeometric test, ****$p < 0.0001$. **c** GO term enrichment of WT (left) and *det2-1⁻/⁻* (right) autonomous seeds treated with auxin plus epi-BR versus treated with auxin only. **d** Comparison of the relative expression of selected genes that are commonly upregulated in WT (left) and *det2-1⁻/⁻* (right) ovules treated with auxin plus epi-BR versus treated with auxin alone. Log₂Fold > 1 and $p > 0.05$. Z-score normalization was performed to center the mean and set the distribution of values to a SD of 1. ****$p < 0.0001$, ***$p < 0.001$, **$p < 0.01$ and *$p < 0.05$.

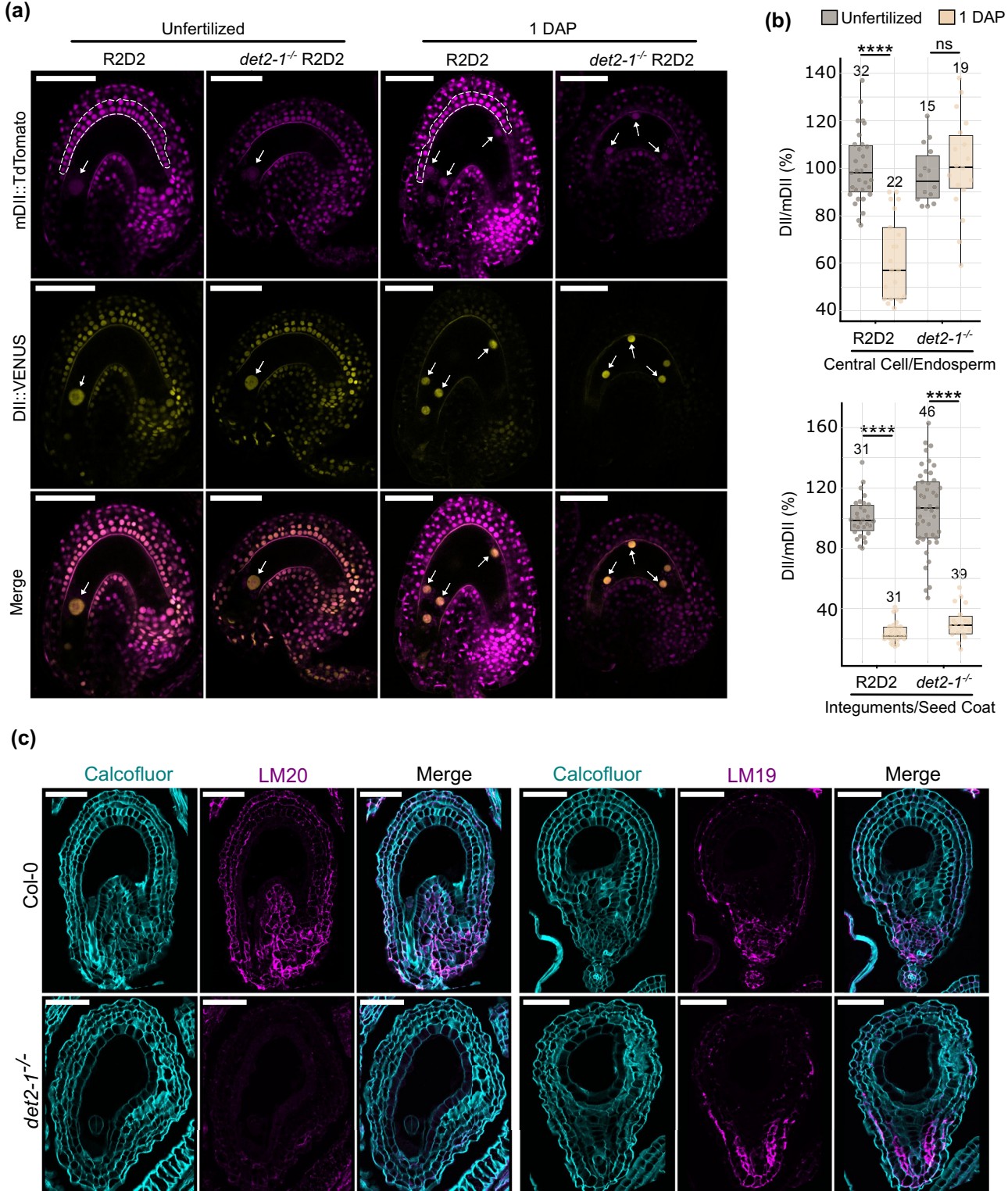

**Fig. 5 | BR mutant seeds have auxin and cell wall-related phenotypes.**
**a** Representative microscopy images of WT and *det2-1⁻/⁻* unfertilized ovules and seeds at 1 DAP expressing the auxin sensor R2D2. Solid arrows indicate the central cell nucleus, and dotted arrows point to endosperm nuclei. Dashed lines surround the nuclei of the integuments and seed coat layers used for quantification. Scale bars, 50 μm. **b** Quantification of DII-Venus/mDII-ntdTomato signal in R2D2 and *det2-1⁻/⁻* seeds before and after fertilization. The values on top of each bar indicate the number of ovules/seeds analyzed. Error bars represent standard deviation. Significance of difference was determined by two-tailed distribution Student's test. ****$p < 0.0001$, ***$p < 0.001$, **$p < 0.01$ and *$p < 0.05$. **c** Labeling of cell wall components in WT (upper row) and *det2-1⁻/⁻* (lower row) at 2 DAP. Cyan is calcofluor white staining and magenta is immunolabelling with either LM19 or LM20 antibodies. Scale bars, 50 μm. These experiments were repeated twice with similar results.

softer cell walls[50]. Indeed, both at 2 and 4 DAP, we observed changes in the cell wall labeling of *det2-1*$^{-/-}$ and *bri1-6*$^{-/-}$ seed coats (Fig. 5c and Supplementary Fig. 7). For instance, there is a clear reduction in the LM20 signal in the cell walls of *det2-1*$^{-/-}$, in comparison to the respective WT. This may signify that *det2-1*$^{-/-}$ seed coats may be stiffer than those of the WT. Overall, the different cell wall composition of seed coats of BR mutants correlates with the transcriptomic data of Fig. 4. We thus theorize that these different cell wall properties may be linked to the reduced seed coat growth observed in BR mutants (Supplementary Fig. 6). If this is true, then the reduced endosperm proliferation also observed in these mutants may be an indirect consequence of their seed coat defects.

### Endosperm proliferation is coupled to seed coat size

Based on the previous observations, we questioned whether the reduced endosperm proliferation in BR mutants could be due to a physical effect imposed by the seed coat. If that is the case, then independent mutations that limit seed size in a sporophytic manner should also exhibit reduced endosperm proliferation. That is, mutations leading to smaller seed coats should phenocopy the endosperm phenotypes we observe in BR mutants. To test this, we used mutants for the general growth regulators KLUH (KLU) and AINTEGUMENTA (ANT), which produce small seeds[46,51]. We confirmed that, indeed, *klu-4*$^{-/-}$, *ant-T*$^{-/-}$ seeds are smaller than the WT at 2 DAP (Fig. 6a). Importantly, these mutants also showed a reduced endosperm nuclei number comparatively to the WT (Fig. 6b), similar to what we observe in BR mutants.

For each of the mutants analyzed in this study, we plotted the average number of endosperm nuclei at 2 DAP vs. seed size at the same time point. We observed an almost perfect linear correlation between the two parameters (Fig. 6c), giving strength to our argument that endosperm proliferation is a function of physical seed coat expansion. Thus, to further assess whether the composition and, hence, the physical properties of the seed coat indeed influence endosperm proliferation, we scored endosperm nuclei number in a mutant for the XYLOGLUCAN XYLOSYLTRANSFERASES 1 and 2 (XXT1/2), which is devoid of cell wall xyloglucans[52]. This mutant also produces smaller seeds compared to the WT and, consistently, less endosperm nuclei (Fig. 6a), linking endosperm proliferation to the physical-chemical properties of the seed coat cell walls. We then tested if exogenously supplied epi-BR would rescue this phenotype, because we showed that BRs modulate the cell wall properties of the seed coats (Fig. 5c). Indeed, exogenous application of 100 µM epi-BR rescued both the size and endosperm nuclei number of *xxt1 xxt2* seeds (Supplementary Fig. 8), supporting our hypothesis.

To further challenge our model, we hypothesized that inhibition of PME activity in a BR mutant background should, to some degree, rescue its endosperm phenotypes. To test this, we treated *bri1-6*$^{-/-}$ and *det2-1*$^{-/-}$ fruits with exogenous epigallocatechin gallate (EGCG), which is a potent inhibitor of PMEs[53]. Indeed, we observed a significant rescue of the *bri1-6*$^{-/-}$ seed size and endosperm nuclei phenotype upon EGCG treatments (Supplementary Fig. 8). Surprisingly, we did not observe the same trend for *det2-1*$^{-/-}$ (Supplementary Fig. 8). We link this to the observation that the seed coat cell walls of *det2-1*$^{-/-}$ are much more depleted in methylesterified pectins, when compared to *bri1-6*$^{-/-}$, and therefore likely considerably stiffer (Fig. 5c and Supplementary Fig. 7). Thus, the effect of EGCG is likely to be less efficient in *det2-1*$^{-/-}$. Interestingly, we also occasionally observed a repressive effect on seed phenotypes when treating WT seeds with EGCG (Supplementary Fig. 8). This fits with previous observations that EGCG applications ectopically increase BR signaling[54], which we have shown is deleterious for endosperm development in excess (Fig. 1a, b and Supplementary Fig. 2).

To further validate that seed coat mechanical properties affect endosperm proliferation by physically restraining endosperm growth, we used a previously described apparatus to compress seeds for 24 h as a way to physically restrict their growth[24]. Strikingly, we observed that seed size and the number of endosperm nuclei were reduced in compressed seeds when compared to control seeds (Fig. 6d), showing that endosperm proliferation can be directly inhibited by physically restraining seed growth. Again, we observed a correlation between the size of the seed, either compressed or not, and the number of endosperm nuclei (Fig. 6e). This data further supports the hypothesis that endosperm development depends on its physical confinement by the seed coat whose wall mechanical properties are influenced by BR.

Finally, because we observed that auxin activity in the endosperm is reduced in BR mutants (Fig. 5a, b), we asked if this reduction could be due to physical compression of the seed cavity. Therefore, we measured R2D2 activity in the endosperms of compressed seeds at 1 and 2 DAP. To achieve this, we created an experimental setup in which the siliques were compressed in between two glass slides attached by adjustable elastics, allowing controlled pressure application to the fruits (Supplementary Fig. 9). At 1 DAP, the seeds of one out of three compressed fruits showed an increased VENUS/tdTomato ratio in comparison to non-compressed seeds, indicating reduced auxin activity in the endosperm (Supplementary Fig. 9). Remarkably, although the differences were not statistically significant ($p = 0.0678$, $t$-test), this was the only silique which showed visible signs of compression (Plant 1 in Supplementary Fig. 9), suggesting that in the remaining tested siliques the pressure exerted was insufficient to induce significant changes. Thus, using the same setup that rendered visibly compressed siliques, we quantified auxin activity in the endosperm of seeds at 2 DAP. At this time, we observed that all plants tested had endosperms with significantly reduced auxin activity (Supplementary Fig. 9), supporting the hypothesis that physical constraints imposed by the seed coat impact auxin activity in the endosperm. Our proposed model can be found in Fig. 6f.

## Discussion

Here, we uncovered a role for BR in endosperm development. Our data shows that mutants impaired in BR biosynthesis and signaling have reduced endosperm proliferation after fertilization and that this is at least in part due to a maternal effect from the seed coat. We then demonstrate that rather than BRs being an actual non-cell autonomous signal, originating in the seed coat and moving to the endosperm, it is the physical constraint imposed by the seed coat that impacts endosperm proliferation (Fig. 6f). Moreover, we show that BRs function in a dose-dependent manner during seed development, as mutants with constitutive BR signaling also show endosperm proliferation defects. This is in line with what has been observed in roots, pollen tubes and in the seed coat, where epi-BR concentrations above a certain threshold are detrimental for growth[20,42,49].

Our reporter analysis showed that the genes involved in BR biosynthesis and signaling are expressed in the integuments and seed coat, but not in the central cell or endosperm. This indicates that BR are produced in the maternal tissues of the seed, where they generate a downstream mechanism to regulate endosperm development. Jiang et al. (2013) suggested a direct role for BR in endosperm development, which would be achieved by direct regulation of *IKU1* and *IKU2* by BZR1[48]. We, however, did not detect *BZR1* expression in the central cell or in the endosperm. In agreement with a sporophytic mode of action, ectopically active BR signaling in the sporophytic tissues of the seed was sufficient to partly rescue endosperm proliferation in the biosynthetic mutant *det2-1*. Reciprocal crosses of *bri1-6* and WT also support this notion. However, we cannot fully rule out a gametophytic or haploinsufficient effect for *det2-1*, or that BR effectors are expressed in zygotic tissues in time points not analyzed here.

BR are hormones with limited mobility[55–57], and recent evidence suggests that BR precursors are transported via plasmodesmata[58], but can also be actively exported from cells by ABCB-type transporters[44].

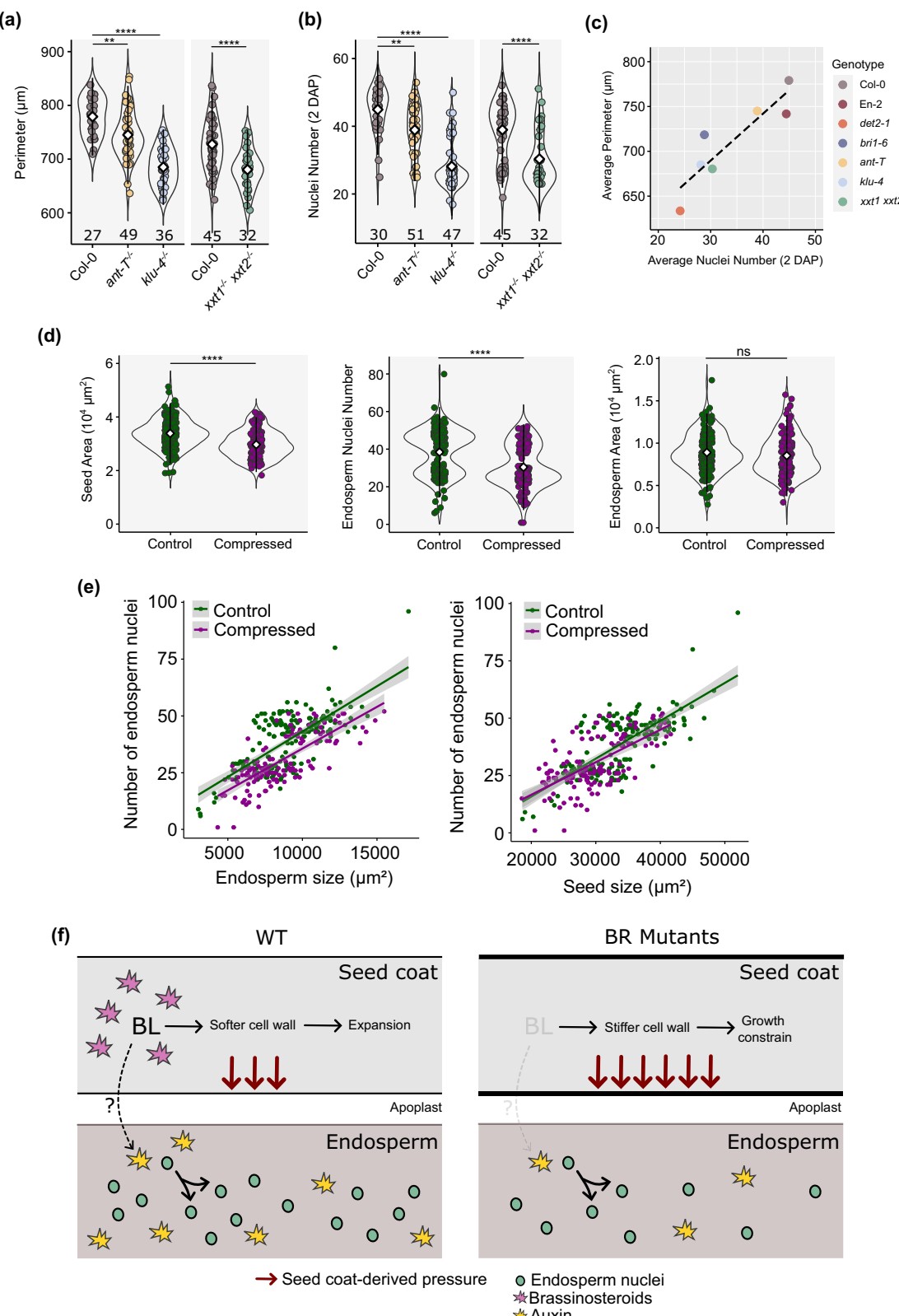

Our results show that the different enzymes required for BR biosynthesis are expressed in distinct domains of the integuments and seed coat. This means that BR precursors must move not only between cells but also between distinct cell layers to achieve the final product. Consistent with this, we found the BR transporters ABCB1 and ABCB19 expressed in several layers of the seed coat. However, given that the inner and outer integuments are predicted to be symplastically isolated[59], it remains to be determined how BR precursors can bypass these barriers.

The timing of endosperm cellularization has been shown to correlate with seed size and endosperm proliferation, two processes that are also linked[12,15]. This led to the hypothesis that the rate of endosperm proliferation, together with the size of the seed coat, determines how much the endosperm grows, by setting the timing of

**Fig. 6 | Restraining seed growth affects endosperm nuclei number. a** Perimeter of *ant-T*[−/−], *klu-4*[−/−] and *xxt1*[−/−] *xxt2*[−/−] seeds comparatively to the respective WT at 2 days after pollination (DAP). Significance of difference was determined by one-way ANOVA. **b** Endosperm nuclei number of *ant-T*[−/−], *klu-4*[−/−] and *xxt1*[−/−] *xxt2*[−/−] seeds comparatively to the respective WT at 2 DAP. Significance of difference was determined by one-way ANOVA. ***$p < 0.0001$, ***$p < 0.001$, **$p < 0.01$ and *$p < 0.05$. Error bars indicate standard deviation. The number below each sample indicate number of seeds analyzed. **c** Correlation between the number of endosperm nuclei and seed size at 2 DAP in different genotypes analyzed in this study. **d** Effect of a 24 h compression of the fruit with a microvice on seed size, endosperm size and number of endosperm nuclei in WT seeds at 2 days after pollination (DAP). Significance of difference was determined using one-way Wilcoxon tests.

****$p < 0.0001$, ***$p < 0.001$, **$p < 0.01$ and *$p < 0.05$. Error bars indicate standard deviation. For the control condition, 177 seeds were analyzed, and for the compressed condition, 130 seeds (two independent experiments). **e** Correlation between the number of endosperm nuclei and seed size, between the number of endosperm nuclei and endosperm size, and between endosperm size and seed size, in WT seeds at 2 DAP whose fruits were compressed for 24 h, as seen in (**d**). Smoothing of the data (colored lines) was done using a linear model. **f** Proposed model. Left, BRs create a permissive environment for seed coat expansion, which impacts endosperm proliferation. The mechanism by which this environment impacts auxin activity remains undiscovered (question mark). Right, loss of BR leads to a stiffening of the seed coat cell walls, which indirectly affects auxin activity in the endosperm, reducing its proliferation rate.

cellularization. Nevertheless, recent reports showed that seed size can, in fact, be uncoupled from the timing of cellularization[18]. And, indeed, our data did show that even though *det2-1* and *bri1-6* seeds are smaller and have reduced endosperm proliferation, the timing of cellularization was indistinguishable from the WT. These observations demonstrate that the onset of cellularization does not necessarily correlate with how much the endosperm proliferates and that the size of the seed coat does not always determine the timing of cellularization. Furthermore, since mutants impaired in BR biosynthesis and signaling have smaller seed coats and endosperms as early as at 2 DAP, yet show normal cellularization, it can be concluded that seed coat size regulates endosperm growth from earlier stages of seed development, even before the cellularization process starts.

Given that BR activity originating in the sporophytic tissues contributes to endosperm development, we hypothesized that this could occur through two distinct mechanisms: (1) activation of a downstream signal, such as another hormone or a peptide, that would move to the endosperm and stimulate its growth; (2) induction of seed coat expansion providing more space for the endosperm to proliferate and grow since the growth of these two tissues occurs in a coordinated manner. This would be consistent with the observation that BR mutant seeds not only have less proliferative endosperms but are also smaller in size. Our transcriptomic data revealed that BR affects the expression of genes that participate in auxin biosynthesis, signaling, metabolism, and transport. Concomitantly, we detected reduced auxin activity in the endosperms of *det2-1*[−/−] and *bri1-6*[−/−], and this likely contributes to their less proliferative endosperms. In agreement with BR stimulating endosperm proliferation, the gene encoding CYCLIN B1;1 (CYCB1;1), which together with CYCB1;2 and CYCB1;3, promote endosperm proliferation[60], was upregulated upon epi-BR treatments (Supplementary Data 1).

BR-mediated cell expansion has also been shown to occur through the regulation of cell wall remodeling processes[22,41]. As such, it is possible that changes in expression of genes involved in cell wall biogenesis and remodeling observed in our transcriptomic dataset could be due to BRs directly regulating the physical properties of the seed coat cell walls. It was previously shown that stiffening of a mechanosensitive seed coat cell wall constrains growth as a response to the turgor pressure exerted by the growing endosperm[18,24]. This model could explain the reduced endosperm and seed size of *iku2* mutants: *iku2* endosperms induce high pressure on the seed coat, which leads to a precocious reinforcement of the mechanosensitive wall, thereby constraining seed growth[18]. In this model, the seed coat "senses" endosperm growth and responds to it by reinforcing its cell walls to resist the growth[18]. It is tempting to hypothesize that the endosperm could also perceive mechanical signals originating in the seed coat, thereby coordinating its growth. In this scenario, the reduced endosperm proliferation observed in BR mutants could be caused by the endosperm perceiving the constraint imposed by the reduced seed coat expansion, thereby slowing down its growth as a response. Indeed, BR mutants accumulate less methylesterified

pectins in the seed coat, revealing that these mutants might have stiffer seed coats compared to the WT. It should be noted that the degree of pectin methyelesterfication was also shown to trigger BR signaling via the RECEPTOR-LIKE PROTEIN 44 (RLP44) that interacts with BRI1 and BAK1[25], indicating that feedback between the cell wall and BR signaling regulates cell growth. By analyzing unrelated mutants that produce smaller seed coats, as well as by physically restraining seed growth, we show that endosperm proliferation is indeed a function of seed coat expansion. Moreover, we show that physical restriction of seed growth has a measurable impact on auxin activity in the endosperm, which we have previously shown is necessary for its proliferation[5]. In our model (Fig. 6f), we thus propose that the physical properties of the seed coat impact auxin activity in the endosperm, thus modulating its proliferation rate. How these physical stimuli from the seed coat are transduced into a molecular output in the endosperm will thus be an interesting new field of research.

## Methods
### Plant materials
The *Arabidopsis thaliana* mutants used in this study were *det2-1*[61], *br6ox1 br6ox2*[42], *cpd-91* (SALK_078291), *dwf4-5D*[34], *bri1-6*[29], *bes1-p*[35], *beh2-1*[62], *BRI1*[OX36], *bzr1-D*[38], *bes1-D*[39], *fie-12*[63], *ant-T* (SALK_022770), *xxt1 xxt2*[64], and *klu-4*[65]. The WT accessions used were Enkheim-2 (En-2) for *bri1-6*, Columbia-3 (Col-3) for *beh2-1* and Columbia-0 (Col-0) for the other mutants. The following published reporter lines were also used: pCPD::NLS::3xEGFP[20], pROT3::ROT3-GFP, pBR6OX1::BR6OX1-GFP and pBR6OX1::BR6OX2-GFP[42], pBRI1::BRI1-GFP[41], pBZR1::BZR1-YFP[43], pBES1::BES1-GFP[42], R2D2[47] and ABCB1::ABCB1:GFP and ABCB19::ABCB19:GFP[66].

Seeds were sterilized with 70% ethanol, followed by a 5-min wash of 5% bleach and 0.1% Triton-X100. Lastly, they were washed with 100% ethanol 2–3 times and air dried. The sterile seeds were plated on ½ MS medium (0.43% MS High Salts Mix, 0.05% MES, 0.8% Plant Agar and 1% sucrose; when necessary, the medium was supplemented with the appropriate antibiotics) and stratified during 48 h at 4 °C in the dark. Plates were then transferred to a growth chamber (22 °C; 16 h light/8 h dark; 50 µmol s⁻¹ m⁻²;70% humidity). After 7–10 days, depending on the genotype, the seedlings were transferred to soil and grown in a phytotron chamber (21 °C; 16 h light/8 h; 150 µmol s⁻¹ m⁻²).

### Developmental assays
The flowers were always emasculated at stages 12–13. For auxin assays, the pistils were dipped in a solution of 100 µM of 2,4-D (Duchefa Biochemie) 2 days after emasculation (2 DAE). The pistils were collected 3 days after treatment (3 DAT). For the BR treatments with variable concentrations, the pistils were first treated with 100 µM of 2,4-D at 2 DAE. On the following day, the pistils were dipped in solutions with 100 pM, 1 µM, 50 µM, 100 µM or 150 µM of epi-Brassinolide (epi-BL) (Sigma). The autonomous siliques were collected 2 days after BR treatment for analysis. For EGCG treatments, the flowers were emasculated, pollinated 2 days later, and 1 day after the pistils were treated with varying

concentrations of EGCG (Sigma). The samples were collected 2 days later. A mock control consisting of water and the same volume of ethanol as used in the hormonal solutions was done in parallel for all experiments. All the solutions were supplemented with 0.01% of Silwet L-77 as a wetting agent. Autonomous endosperm formation was tested by emasculating the pistils and collecting them 5 DAE. For fertilization assays, the pistils were hand-pollinated 2 DAE. The samples were collected 2 days after pollination (2 DAP). The imaging of the reporters was done at 1 DAP.

### Histological and fluorescence analysis

For seed clearings, the pistils and/or siliques were fixed in ethanol:acetic acid (9:1) overnight. The samples were then washed with 90% ethanol for 10 min, followed by another 10-min wash with 70% ethanol. The samples were then cleared overnight in chloralhydrate solution (66.7% chloralhydrate (w/w), 8.3% glycerol (w/w)). The samples were dissected in the same solution on a glass slide under a Leica S9E binocular. The ovules/seeds were observed under differential interference contrast (DIC) optics using an Olympus BX51 microscope. Images were taken using an Olympus DP74 camera.

For fluorescent analyses, the samples were mounted in 0.1 mg/mL propidium iodide (PI). The fluorescent lines were analyzed with the Leica STELLARIS 8 DIVE confocal microscope using the following settings (in nm; excitation-ex and emission-em): vYFP—ex 514, em 519–577; mDII:tdTomato (RFP)—ex 561, em 599–622; and GFP—ex 488, em 499–525.

For Feulgen staining of seeds, the siliques were opened and fixed in ethanol:acetic acid (3:1) overnight. Then, the samples were washed in water three times for 15 min each wash and incubated in 5 N HCl for 1 h. Similar to the first wash, the samples were washed with water three times, each for 15 min. The washed siliques were next incubated in Schiff's reagent (Clin-Tech) for 4 h. Afterward, the siliques were washed three times with cold water, with each wash taking 10 min, followed by a series of 10-min washes with 10%, 30%, 50%, 70%, 95% and 99.5% ethanol. The 99.5% ethanol wash was repeated several times until the solution came out colorless (overnight incubation and one day with 2–3 h intervals between ethanol changing). The samples were posteriorly incubated in 99.5% ethanol:LR white resin (3:1) and 2:1 for 15 min each incubation, followed by 1:1 for 1 h. After that, the siliques were incubated overnight in LR White resin supplemented with a polymerization catalyst (Agar Scientific). Then, the seeds were mounted on LR White and baked overnight at 60 °C. The material was analyzed under the Leica STELLARIS 8 DIVE multiphoton microscope with emission at 488 nm and excitation at 490–622 nm.

For endosperm nuclei counting and R2D2 quantification in the central cell and endosperm, images were analyzed using ImageJ (https://imagej.net/). The R2D2 quantification in the central cell and endosperm was done by calculating the ratio between signal intensities from the YFP and RFP channels after removing the background noise. To achieve this, the central cell or endosperm nuclei were marked and the YFP and RFP fluorescent signal measured. The mark was then moved to the proximity of the central cell or endosperm nuclei to measure the background noise of both channels. For endosperm nuclei, the fluorescent signal of two or more nuclei was collected from each image, and the average ratio was calculated. To quantify R2D2 in the integuments and seed coat, images were analyzed using Imaris x64 9.1.2 (Oxford Instruments). The nuclei from the two innermost cell layers of the inner integuments/seed coat were detected using the spot function (Fig. 5a, dashed lines). The estimated XY diameter varied between 3.20 and 3.50 μm according to the size of the nuclei observed for each genotype and condition (unfertilized versus fertilized). The nuclei detected outside of the inner integument/seed coat layers or out of the focus plane were manually removed. In addition, nuclei that were previously undetected by the spot function were manually selected. The nuclei detection/selection was performed

in the mDII:tdTomato channel. The option "background subtraction" was selected to automatically remove the background signal from the YFP and RFP intensity values. The average YFP and RFP intensities and respective standard deviations were determined for each ovule and seed and the ratio between the YFP and RFP calculated.

### Seed compression

To physically restrict seed growth, fruits at 1 day post-anthesis, still attached to the plant, were compressed using a microvice as previously described[21]. After 24 h, control and compressed fruits (at 2 days post-anthesis) were harvested and the seeds were cleared and imaged by differential interference contrast (DIC) microscopy as described in the previous sections.

To quantify auxin activity in compressed seeds, pistils were hand-pollinated at 2 DAE and the fruits compressed 6 h and 24 h after pollination for 1 and 2 DAP assays, respectively. Control and compressed fruits were imaged 24 h after pressure application as described in the section above.

The resulting images were analyzed using a specific macro for FIJI software and are available at: https://github.com/RDP-vbayle/SiCE_FIJI_Macro/tree/main/misc. Briefly, this macro concatenates the different images of the same seed (imaged by DIC microscopy with different focuses) into a single stack. Seed and endosperm ROIs (Regions Of Interest) are manually defined using the polygon selection tool and measured after Spline fitting. Afterward, endosperm nuclei are identified on the proper focal planes using the multipoint selection tool. All results, including embryo stage, are then summarized in a result table.

### Immunolabelling

Cell wall immunolabeling was performed as described previously[67,68]. Briefly, seeds were harvested and fixed using FAA (Formalin–Acetic Acid–Alcohol Solution), followed by serial dehydration steps in ethanol that was gradually replaced with Histoclear using an automated sample preparation machine (Leica ASP300S). The samples were embedded in paraffin, sectioned (4 μm thickness) and dewaxed. The sections were probed using primary antibodies LM19 and LM20 (Agrisera) and labeled with antibody Alexa 647 (ThermoFisher) and counterstained with Calcofluor. Labeled samples were imaged using a Leica SP5 confocal microscope equipped with a 20× objective. Alexa 647 and Calcofluor was excited using a 642 nm and 405 nm laser, with emission recorded between 660–760 nm and 415–500 nm, respectively.

### Molecular cloning and generation of transgenic plants

The promoters of *BRI1*, *BAK1* and *DET2*, as well as the coding region of *BRI1*, were amplified using genomic DNA from Col-0 as template. The primer sequences can be found in Supplementary Table 1. The amplified fragments were purified from gel and recombined into a donor vector (pDONR221) to generate entry clones using Gateway technology. Upon entry clone sequencing, pBRI1::GFP, pBAK1::GFP and pDET2::GFP reporters were created by recombining *BRI1*, *BAK1* and *DET2* promoters into pB7WG2.0 (VIB, Ghent). The *pKLU::BRI1* construct was generated by recombining the *BRI1* coding region into a vector containing the *KLU* promoter[69]. The constructs were then transformed into *Agrobacterium tumefaciens* strain GV3101, which was used to transform *Arabidopsis thaliana* plants by floral dip[70]. The transformants were selected using appropriate antibiotics.

### Statistical analyses

*Microsoft Excel* and *R Studio* were used for data analysis. Significance of differences was determined with the chi-squared test for autonomous endosperm formation and one-way ANOVA or Wilcoxon tests for sexual endosperm quantification. For multiple sample comparison, ANOVA was followed by Tukey's HS multi-comparison test. Samples were considered statistically different when $p < 0.05$. ****$p < 0.0001$, ***$p < 0.001$, **$p < 0.01$ and *$p < 0.05$.

## RNAseq analysis

Col-0 and *det2-1* pistils were dipped in a solution containing either 100 μM of 2,4-D or 100 μM of 2,4-D and 100 μM of epi-Brassinolide (epi-BL) (Sigma). The solutions were supplemented with Silwett L-77 as wetting agent. The autonomous seeds were harvested 3 DAT in RNA*later* (Invitrogen). For each genotype and treatment, three biological replicates were generated. Total RNA was extracted using the MagMAX™ Plant RNA Isolation Kit (Thermo Fisher Scientific), and the mRNA was isolated using the NEBNext® Poly(A) mRNA Magnetic Isolation Module (New England Biolabs), following the manufacturer's instructions. Sequencing libraries were generated using the Collibri™ Stranded RNA library Prep Kit for Illumina™(Thermo Fisher Scientific). The libraries were sequenced in BGI Tech Solutions (Hong Kong) on a MGISEQ-2000 platform, using 100-bp paired-end reads. The reads were mapped to the *Arabidopsis* (TAIR10) reference genome after index trimming using CLC Genomics Workbench (QIAGEN). Only two biological replicates were analyzed for Col-0 treated with 2,4-D and epi-BL since a PCA plot generated in CLC Genomics Workbench indicated that one of the replicates did not cluster properly with the others (Supplementary Fig. 4). Differential gene expression was performed using DEseq[70]. For the differential gene expression analysis, only genes with 10 or more counts per million were analyzed. Genes were considered differentially expressed when log2Fold > 1 (upregulated genes) or log2Fold < −1 (downregulated genes) and false discovery rate (FDR) *p*-value < 0.05. Gene Ontology (GO) enrichment was performed using clusterProfiler[71]. The heatmap was constructed using the average of the normalized counts for each genotype and condition.

## Reporting summary

Further information on research design is available in the Nature Portfolio Reporting Summary linked to this article.

## Data availability

The RNAseq datasets used are deposited in NCBI under reference PRJNA1041724. Source data are provided with this paper.

## Code availability

ImageJ macro for counting endosperm nuclei [https://github.com/RDP-vbayle/SiCE_FIJI_Macro/blob/main/misc/Endosperm-Nuclei-Lima-et-al-2023.ijm].

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

## Acknowledgements

We thank Kerstin Zander and Marcos Martín-Sanchez for technical assistance. We thank Jenny Russinova for providing several reporter lines for BR biosynthesis genes as well as BR mutants, Jurgen Kleine-Vehn for the BRI1 reporter, Stefanie Sprunck for the CPD reporter, Juthamas Chaiwanon for *det2*, Sunghwa Choe for *dwf4-5D*. We also thank Kian Hématy, Juthamas Chaiwanon and Kamil Ruzicka for additional reporter lines, which finally were not included in the manuscript. This work was funded by project number 421178202 of the German Research Foundation (DFG) and by the Max Planck Society.

## Author contributions

R.B.L., R.P., B.L., A.S. and D.D.F. designed the experiments. R.B.L., R.P., S.T.E., P.F., A.F., B.L. and D.D.F. performed the experiments. V.B. designed the script. R.B.L., S.T.E., B.L., A.S. and D.D.F. analyzed the data. R.B.L. and D.D.F. wrote the manuscript with input from all co-authors.

## Funding

## Competing interests

The authors declare no competing interests.
