## [Transparent Peer Review file · Nature Communications]

Seed coat-derived brassinosteroid signalling non-cell autonomously contributes to endosperm development

Corresponding Author: Dr Duarte Figueiredo

Version 0:

Reviewer comments:

Reviewer #1

(Remarks to the Author)

In this manuscript Lima and coauthors analyzed endosperm development in an array of BR biosynthesis and signaling mutants. They determined the expression pattern of BR biosynthesis and signaling genes. Through transcriptomics analysis, they revealed that BR mutant seeds were affected in cell wall and auxin-related processes. They then attempted to prove that the auxin signaling and physical property of seed coat cell wall act downstream of BR's effect on endosperm development. In general, it is a timely work and provides new perspective in the regulatory network of endosperm. However, the manuscript could be improved in writing so that readers who are not experts in BR field could easily follow the story. There are also several points need to be strengthened.

My comments/suggestions are as followed:

Lines 89-97, Fig. S1: Only one ovule/seed for each genotype or treatment was shown. A quantitative analysis is required. An explanation of why these observations indicate that "a seed coat-derived signal would be involved in promoting endosperm proliferation in a non-cell autonomous manner" would be helpful for readers.

Lines 103-105, Fig. S2: As not everyone who reads the paper is familiar with BR biosynthesis/signaling pathway, a short introduction of these genes would be necessary for readers to understand what the mutants are. This applies to other genes/mutants when they are mentioned for the first time.

Fig. 1a, 1d, S3b: the nuclei number is reduced in all the three BR biosynthesis mutants, *det2-1*, *br6ox1/2* and *cpd-91* mutants, but the rate of autonomous seed formation was different in these mutants, reduced in *det2*, unchanged in *cpd-91*, and increased in *br6ox1/2*. The nuclei number is unchanged in *bzr1-2*, *bes1-p* and *beh2-1*, but again the frequency of endosperm with ovules are oppositely in these mutants, increased in *bzr1-2* and *bes1-p*, but decreased in *beh2-1*. How could these data be interpreted?

I noticed a big variation for the ovule frequency between the two Col-0 in Fig. 1d. This variation is equal or even bigger than those between genotypes/treatment, for example, Col-0 and *br6ox1/2*, En-2 and *bri1-6*, as well as between the mock and auxin/eBL treatment in Fig. 1g, which the authors have determined as significant difference. This raises the concern if the difference between genotypes/treatment is true or just from technical errors. More replicates might solve this issue. Similar concern also points to Fig. S3b.

Fig. 3a and S5: As the phenotypes in Fig. 1 are analyzed at 2DAP, I am wondering why are the reporters not analyzed at the similar experimental conditions. An explanation for why 1DAP was selected in the manuscript would be sufficient.

Lines 228-229: What is the evidence for the conclusion that BR produced in the seed coat is important for endosperm development? Could the endosperm defects of *det2-1* be recovered by the expression of DET2 in seed coat? If so, this conclusion is convincing.

Fig. 4a: it is unclear which two groups are compared for the ns.

Fig. 5a,b: The fluorescence of DII::VENUS is absent in the *det2-1* endosperm and seed coat at 1DAP. How is the ratio of

DII/mDII analyzed?

Lines 265-269, Fig. S7: in Fig. S7a, the DII::VENUS and mDII::TdTomato fluorescence are similarly changed in central cell / endosperm of the control and bri1-6 mutants, which does not fit to the quantification data of DII/mDII in central cell/endosperm in Fig. S7b.

Fig. 5c and S9: Do bri1-6 mutants display similar change in pectin methyl esterification in the seed coat? In addition, a quantification analysis would strengthen these data.

Lines 310-314, Fig. S10: Does EGCG have similar effect on det2-1 mutants as it does for bri1-6? Why are some experiments done with bri1-6 and others with the bri1-6 mutants?

Lines 314-317: the interpretation for the observation in EGCG-treated WT seeds seems not fit to the working model. If the physical property of seed coats acts as downstream of BR's effect on endosperm, the effect of EGCG on cell wall will determine seed size but not the upstream BR signaling.

Lines 320-323: the description for endosperm size is not consistent with the data shown in Fig. 6d. Fig. 6d showed that the endosperm area after compression is comparable to that of the control, but not reduced.

Fig. 6: Could compression inhibit the effect of BR in promoting auxin-induced endosperm formation? Similarly, how is the effect of exogenous application of BR on the cell wall mutants like xxt1/2? These need verification if the authors want to claim that physical confinement acts downstream of BR in regulating endosperm proliferation and size.

For the working model: what is the relationship between auxin and mechanical pressure? Are they in the same pathway or independent factors? Will compression impact on auxin activity?

Reviewer #2

(Remarks to the Author)

The development of embryo, endosperm and integument synergistically influences seed development and determine seed size and shape. Embryo and endosperm are zygotes but integument (seed coat) is maternal tissue. The communications among these tissues are important for the seed development, which regulatory mechanisms remain unclear. This manuscript entitled "Seed coat-derived brassinosteroids non-cell 3 autonomously regulate endosperm development" revealed that BR regulates cell wall-related processes in the seed coat, and that the biophysical properties of this maternal organ determine the proliferation rate of the endosperm in a manner independent from the timing of its cellularization. And the authors proposed that BR signaling in maternal tissues tunes endosperm proliferation to seed coat expansion. The topic is interesting, and the manuscript shows lots of work. However, there isn't a main thread for the readers to follow, and the logicity is not clear enough. It is difficult to understand the data and conclusion. The organization of this manuscript needs to be significantly improved. And some conclusions should be turn down to avoid overstatement. Taken together, this manuscript could not be considered for publishing in the Nature Communications unless substantial modification is made, and all the questions raised by reviewers are addressed. The main concerns (not all) are listed below.

Major points:

1. The authors used fertilization-independent (autonomous) endosperm to study the communications between endosperm and integuments, which would be artificial and could not reflect the real conditions. According to that, the conclusion should be based on this specific condition but not native condition.
2. The treatment of BR and auxin in BR-deficient mutant should be very careful. Since BR treatment could complement the phenotype of BR deficiency, there would not be significant difference between the BR and auxin treatment in det2 and auxin treatment in WT, unless the special design of treatment way. Besides, BR and auxin crosstalk a lot in different levels. It is necessary to exclude the indirect BR influences in auxin regulation of embryo/endosperm/integument development.
3. The authors detected the expressions of many components of BR signaling and concluded that BR related genes didn't express during endosperm development. But in figure s1, BZR1 was expressed in embryo. In addition, the data from Arabidopsis eFP browser showed that DET2 and BZR1 were expressed in the embryo. Besides, the authors did not detect BZR1 expression in the central cell or in the endosperm. However, the data from Arabidopsis eFP browser showed that BZR1 was expressed in the endosperm. Based on the unpublished data from the collaborators and I, at least BZR1 clearly expressed in embryo sac including central cell (might not in nuclei). Probably BR is existing and playing roles in endosperm, and BZR1 regulated genes might still work during endosperm development. At least, the possibility could not be excluded. I will show the pictures if necessary. In Summary, the authors need to provide additional data to determine whether BZR1 was expressed in the endosperm or not. Otherwise, the conclusion that "BR regulate endosperm development via the maternal sporophytic tissues" is overstate.
4. One of important conclusions is "BR regulate cell wall-related processes in the seed coat, and that the biophysical properties of this maternal organ determine the proliferation rate of the endosperm in a manner independent from the timing of its cellularization", which based on the endosperm cellulation time of det2-1 and bri1-6. Basically, this conclusion would be right, but needs to be limited. The research of endosperm cellulation proposed that the later endosperm cellulation, the larger seeds. The authors cladded that the BR deficient mutants have smaller seeds but no significant different endosperm cellularization time (the time relative to embryo development, but not the day after fertilization since det2-1 develops generally slowly than WT), which is consistent to the previous publications (Jiang et al., 2013). However, BR regulates the expression of SHB1, MINI3 and IKU2, and the genetic experiments illustrated these genes work downstream of BZR1 (Jiang

et al., 2013). Although it is unclear that why BR regulates the expression of genes related to endosperm cellularization and seed size but did not affect the endosperm cellularization time, the reason that authors mentioned for not supporting the direct BR regulation of endosperm development (BZR1 didn't express in central cell or endosperm) might not be solid enough (see major point 2). Furthermore, endosperm cellularization is still an important regulatory mechanism of seed size. I suggested the authors to carefully discuss this part and turn down the conclusion to something like "our results revealed that BR regulates the proliferation rate of the endosperm through maternal way (cell wall-related processes in the seed coat), and we could not exclude that other ways..."

5. In the result 1, the *det2-1* and double mutant *br6ox1/2* seeds produce fewer endosperm nuclei compared to WT. A reduction in the rate of autonomous seed formation was observed in the *det2-1* mutant, but not observed in the double mutant *br6ox1/2*. Since DET2 and BR6OX1/2 are key enzymes of BR synthetic pathway, the authors should explain what mechanism led to this difference.

6. For the endosperm nuclei phenotype, *det2* (BR deficient mutant) and *bes1-D/bzr1-D* (BR signal enhanced mutant) led to similar phenotype. The authors clade that the decrease in nuclei number observed in *bzr1-D* is likely caused by excessive signaling rather than low endogenous BR. However, they saw no differences between *bzr1-D/-dwf4-5D/-* and the single mutant *bzr1-D/-*, indicating there isn't feedback inhibition of BR synthesis. Thus, how can the authors explain the reason that BR deficient mutant and BR signal enhanced mutant led to similar phenotype? It seemed that BR probably did not play direct roles in this process? These results make the readers confused because they are different from many BR-regulated biological functions published previously. The authors should provide additional data to demonstrate this point.

Minor points:

1. The descriptions like "We hypothesized that... This was based on..." and "We then hypothesized that... Therefore, we analyzed..." didn't match the normal reading habits of research article and has strong potential assumption.
2. There are too many clauses in this manuscript, which made sentences too long and difficult to understand.
3. The figures are a little bit preliminary. It is much helpful to label important parts in the pictures, such as the endosperm nucleus in Fig1b and c.
4. The figure legends are too simple. The detailed information is needed to understand the results, such as the method of statistical analysis.

Reviewer #3

(Remarks to the Author)

This manuscript by Lima et al., reports that seed coat-derived brassinosteroids non-cell autonomously regulate endosperm proliferation. They suggest that BR does so by affecting auxin signaling and by regulating cell-wall softening. This is an interesting finding; however, the current data are not sufficient to support the main conclusion. My comments are listed below:

1. If BR is required for endosperm proliferation, the mutations resulting constitutive BR signaling should lead to increased endosperm proliferation. However, *bes1-D* and *bzr1-D* actually showed a reduction of endosperm proliferation. These observations do not support the author's conclusion and should be carefully interpreted.
2. Genes involved in BR perception, biosynthesis and TFs are expressed in the endosperm (Fig S1). Thus, BR biosynthesis in the endosperm might impact endosperm development. So the observed reduction of endosperm proliferation in BR mutants might not because of lack of cell coat-derived BR.
3. BR mutants have reduced auxin activity in the endosperms, however, no evidence was provided to show that it is the reduced auxin activity that leads to decreased endosperm proliferation.
4. BR mutants have stiffer seed coats. However, no data was provided to support that stiffer seed coats contribute to the reduced endosperm proliferation. Cell wall softening mutants might be used to cross with BR mutants and then check the endosperm proliferation phenotype. Several unrelated mutants that produce smaller seed coats have reduced proliferation; however, because they are selected; it is still possible that other mutants with smaller seed coats might have normal endosperm proliferation.

Version 1:

Reviewer comments:

Reviewer #1

(Remarks to the Author)

The authors have addressed all my concerns.

Reviewer #2

(Remarks to the Author)

The revised version has been improved in logicity and organization. Some points became clearly. However, they remove most data related to auxininduced autonomous seeds from the manuscript, leading to reduced depth of mechanism and

innovation.

Furthermore, the author wrote the hypothesis first, then described the results later, which gives readers a impression that the author presumed the conclusions and subsequently demonstrated them. In my opinion, it would be better to modify the descriptions.

Reviewer #3

(Remarks to the Author)

The authors made some changes in response to my previous concerns. However, the conclusion "BR regulate endosperm development via the maternal sporophytic tissues" and the model in figure 6f remain seriously misleading. The authors have not adequately address whether BR exists and plays a role in tissues other than the seed coat, such as endosperm. Data in Figure 3d and 3f showed that reconstruction of BR signaling in the seed coat only marginally rescued of the *det2-1* and *bri1-6* endosperm phenotypes, indicating that BR signaling in tissues other than the seed coat is playing even more essential roles in endosperm development than seed coat does. Consistent with this notion, the gene expression data showed that BR pathway genes are expressed in endosperm, as also raised by reviewer 2. The title should be revised to avoid implying that BR affects endosperm development exclusively via the seed coat.

The authors suggested a dose-dependent mechanism to explain why constitutive BR signaling caused unexpected reduction of endosperm proliferation. However, this explanation contradicts findings from the *BRI1* overexpression line, which did not show reduced endosperm proliferation. These conflicting results do not support the authors' conclusions.

In sum, I recommend the authors be careful when interpreting their data and to tone down their conclusions throughout the manuscript.

We thank all reviewers for the careful evaluation of our manuscript. Below we provide a point-by-point response to your comments. Major modifications to the text are indicated in red. We also reorganized the figures to include new data and to better convey our conclusions.

Reviewer #1 (Remarks to the Author):

In this manuscript Lima and coauthors analyzed endosperm development in an array of BR biosynthesis and signaling mutants. They determined the expression pattern of BR biosynthesis and signaling genes. Through transcriptomics analysis, they revealed that BR mutant seeds were affected in cell wall and auxin-related processes. They then attempted to prove that the auxin signaling and physical property of seed coat cell wall act downstream of BR's effect on endosperm development. In general, it is a timely work and provides new perspective in the regulatory network of endosperm. However, the manuscript could be improved in writing so that readers who are not experts in BR field could easily follow the story. There are also several points need to be strengthened.

Thank you for your encouraging words and for the constructive criticism. Your suggestions very much helped improve our work. We agree that the manuscript was not easily accessible to those outside of the BR field. We have made a few modifications throughout the text to make it accessible for a broader audience. We also added new data, as you suggested, which we believe strengthens our claims.

My comments/suggestions are as followed:

1. Lines 89-97, Fig. S1: Only one ovule/seed for each genotype or treatment was shown. A quantitative analysis is required. An explanation of why these observations indicate that “a seed coat-derived signal would be involved in promoting endosperm proliferation in a non-cell autonomous manner” would be helpful for readers.

We now include a quantification of these phenotypes and moved the data to Fig. 1, as this is an important point in our manuscript.

We added a more clear explanation why we proposed that a seed coat derived signal should promote endosperm proliferation. See lines 90-106.

2. Lines 103-105, Fig. S2: As not everyone who reads the paper is familiar with BR biosynthesis/signaling pathway, a short introduction of these genes would be necessary for readers to understand what the mutants are. This applies to other genes/mutants when they are mentioned for the first time.

We agree that it is challenging for readers outside of the BR to keep track of all gene names.

We now added a simplified diagram of the BR biosynthesis and signalling pathway to Fig. 1g, so that readers can easily refer back to.

3. Fig. 1a, 1d, S3b: the nuclei number is reduced in all the three BR biosynthesis mutants, *det2-1*, *br6ox1/2* and *cpd-91* mutants, but the rate of autonomous seed formation was different in these mutants, reduced in *det2*, unchanged in *cpd-91*, and increased in *br6ox1/2*. The nuclei number is unchanged in *bzr1-2*, *bes1-p* and *beh2-1*, but again the frequency of endosperm with ovules are oppositely in these mutants, increased in *bzr1-2* and *bes1-p*, but decreased in *beh2-1*. How could these data be interpreted?

Indeed the autonomous seed data is not easy to interpret. We assume that different BR intermediates may have functions which are specific for asexually produced endosperms, which is not reflected in their sexual counterparts. However, this is speculative. We agree that the artificial nature of these datasets limits the conclusions that we can draw from them. Because these assays caused confusion to several reviewers, we decided to restructure the manuscript and focus on sexually produced endosperms. We removed most data related to auxin-induced autonomous seeds from the manuscript, leaving only that which is necessary to understand the design of the RNAseq experiment.

4. I noticed a big variation for the ovule frequency between the two Col-0 in Fig. 1d. This variation is equal or even bigger than those between genotypes/treatment, for example, Col-0 and *br6ox1/2*, *En-2* and *bri1-6*, as well as between the mock and auxin/eBL treatment in Fig. 1g, which the authors have determined as significant difference. This raises the concern if the difference between genotypes/treatment is true or just from technical errors. More replicates might solve this issue. Similar concern also points to Fig. S3b.

Indeed, we do see some variation when inducing asexual seed formation with auxin. We know that this phenotype is dependent on environmental conditions, so it is sufficient that the experiment is done in different growth chambers for the penetrance of the phenotype to be different in Col-0. However, we do see consistent results between the WT and mutants. We did perform these experiments multiple times, and the mutants always behave consistently, relative to the WT.

However, as mentioned in the response to point 3, we decided to remove most data on asexual seeds from the manuscript, as it was causing confusion to several reviewers, and did not really add much to our conclusions. We now focus on sexual endosperms. This makes for a more cohesive story.

5. Fig. 3a and S5: As the phenotypes in Fig. 1 are analyzed at 2DAP, I am wondering why are the reporters not analyzed at the similar experimental conditions. An explanation for why 1DAP was selected in the manuscript would be sufficient.

The reason why we analysed the reporters at an earlier timepoint was because we expected those genes to be expressed before the phenotypes were detectable, i.e., at 2 DAP. We now made this clear in the manuscript (line 215).

6. Lines 228-229: What is the evidence for the conclusion that BR produced in the seed coat is important for endosperm development? Could the endosperm defects of *det2-1* be recovered by the expression of *DET2* in seed coat? If so, this conclusion is convincing.

We agree that this is an important point. We generated lines expressing DET2 specifically in the seed coat of *det2* mutants using the sporophytic promoter of KLU. Indeed, we observed a partial but significant rescue of the *det2* endosperm phenotype in several independent lines in the T1 generation. However, in several transgenic lines this rescue was not recapitulated in the T2. Our best interpretation was that there were multiple T-DNA insertions in the T1, which segregated in the T2 and the effect was diluted. We are happy to show this data to the reviewer, but decided against including the results in the manuscript, because of the inconsistency of results between generations.

As an alternative we also generated KLU::BRI1 constructs and used them to complement the *bri1* mutant (a reminder that the KLU promoter is specific to the seed coat tissues). As hypothesized we saw a rescue of the *bri1* phenotypes in the complementation lines. This links BR activity in the seed coat to endosperm proliferation. We added this data to Fig. 3.

We thus now mention throughout the manuscript that BR signalling via BRI1 in the seed coat is what regulates endosperm proliferation. And are more careful about claims regarding the site of BR production.

Nevertheless, we believe we provide several pieces of evidence that BR activity in the seed coat determines endosperm proliferation rates: 1) all BR machinery is specifically expressed in the seed coat; 2) BR mutant phenotypes are strongly maternal in origin; 3) *det2* endosperm phenotypes can be rescued by ectopic BR activation in the sporophyte, and *bri1* phenotypes are rescued by sporophytic BRI1 expression.

7. Fig. 4a: it is unclear which two groups are compared for the ns.
It is fixed.

8. Fig. 5a,b: The fluorescence of DII::VENUS is absent in the *det2-1* endosperm and seed coat at 1DAP. How is the ratio of DII/mDII analyzed?

The DII::VENUS signal in *det2-1* at 1 DAP is actually quite strong (indicated by the arrows), so we believe this is a misunderstanding. Regarding the seed coat signal, it is indeed very weak, but still present and detectable using the IMARIS software (Oxford Instruments). We are providing the raw measurements in the separate "Source Data" file.

9. Lines 265-269, Fig. S7: in Fig. S7a, the DII::VENUS and mDII::TdTomato fluorescence are similarly changed in central cell / endosperm of the control and *bri1-6* mutants, which does not fit to the quantification data of DII/mDII in central cell/endosperm in Fig. S7b. If one looks at the merged channels (panels at the bottom), it can actually be seen that the *bri1* endosperm nuclei look "more yellow" than the WT ones. However, it should be noted that we measured fluorescence on endosperm nuclei of 20-30 seeds per genotype and timepoint, so some variation is expected. As mentioned in the response to point 8, we now provide the raw measurements in the "Source Data" file. We can also provide the original microscope pictures upon request.

10. Fig. 5c and S9: Do *bri1-6* mutants display similar change in pectin methyl esterification in the seed coat? In addition, a quantification analysis would strengthen these data.

As suggested, we now did the same experiment with *bri1*. The data was added to Fig. S7. Regarding the quantification of the signals, we do see your point, but we do not find this to be very informative. This would make sense if we were comparing the signal within a specific cell layer, for example. But in our case we see large fluctuations of the signals within single seeds. For example, LM19 fluorescent is relatively weak in the seed periphery, but very strong in the chalaza region (to the point that the signal is likely saturated). So a global quantification can be misleading.

Nevertheless, we are now more careful about the interpretations of these results. Our transcriptomic data points to cell walls of BR mutants being somewhat different from their WT counterparts. With the immunolabelling we simply wanted to validate these results and see if indeed this hypothesis was sound. We realize we cannot make strong statements about rigidity or softness simply based on the methylesterification status of pectins. We now made changes in the manuscript to make this clear: BR affects cell wall properties, which are likely linked to how much the seed coat can expand, which in turn influences how much the endosperm can proliferate.

11. Lines 310-314, Fig. S10: Does EGCG have similar effect on *det2-1* mutants as it does for *bri1-6*? Why are some experiments done with *bri1-6* and others with the *bri1-6* mutants? As suggested, we now also did the EGCG treatments in *det2* (Fig. S8). However, while the assays with *bri1* are quite consistent, we did not see the same trend for *det2*. We link this to the fact that *det2* cell walls are much less rich in methylesterified pectins, when compared to *bri1*. Therefore, in order for EGCG to have an effect it likely needs comparatively more time or higher concentrations. Because we directly apply the chemical to the silique, and not the seeds, we cannot control how much of it actually goes inside. This does not seem to be an issue for *bri1*, where likely small amounts of EGCG are enough to trigger an effect, but for *det2* this is insufficient.

12. Lines 314-317: the interpretation for the observation in EGCG-treated WT seeds seems not fit to the working model. If the physical property of seed coats acts as downstream of BR's effect on endosperm, the effect of EGCG on cell wall will determine seed size but not the upstream BR signaling.

We now realize we did not make this point clear enough in the manuscript. We indeed propose that BR modulate the physical properties of the seed coat cells walls, and thus non-cell autonomously regulating endosperm proliferation. With the EGCG assays, however, we expected a phenocopy of the effect of BR. That is, if seed coat cell wall properties are involved in determining the proliferation rate of the endosperm, then altering those properties (irrespective of BR) should be sufficient to modulate endosperm proliferation. We made this more clear in lines 403-418.

13. Lines 320-323: the description for endosperm size is not consistent with the data shown in Fig. 6d. Fig. 6d showed that the endosperm area after compression is comparable to that of the control, but not reduced.

You are correct. Even though there seems to be a reduction in endosperm size upon pressure, the differences are not significant. We corrected this in the text.

14. Fig. 6: Could compression inhibit the effect of BR in promoting auxin-induced endosperm formation? Similarly, how is the effect of exogenous application of BR on the cell wall mutants like *xxt1/2*? These need verification if the authors want to claim that physical confinement acts downstream of BR in regulating endosperm proliferation and size.

Unfortunately the compression experiments do not work well in the auxin-induced endosperm formation, as the unpollinated pistils are quite fragile and are easily damaged with pressure.

However, as requested, we did perform BR treatments on the *xxt1/2* double mutant. As hypothesized, these treatments rescue the seed coat and endosperm phenotypes of the mutant. We added this data to Fig. S8.

15. For the working model: what is the relationship between auxin and mechanical pressure? Are they in the same pathway or independent factors? Will compression impact on auxin activity?

This is an important point. To test it, we repeated the compression experiments in plants expressing the R2D2 auxin reporter. And, indeed, we observed that seeds under pressure have less auxin activity in the endosperm. This links exogenous mechanical signals to auxin activity and to endosperm proliferation, as we have previously shown that auxin signalling is necessary for endosperm proliferation (Figueiredo et al, 2015, Nature Plants). We now added this data to Fig. S9.

Of course, this raises the question on how these mechanical stimuli non-cell autonomously regulate auxin activity in the endosperm. We do not have the answer to this yet, but it will be the subject of future research.

Reviewer #2 (Remarks to the Author):

The development of embryo, endosperm and integument synergistically influences seed development and determine seed size and shape. Embryo and endosperm are zygotes but integument (seed coat) is maternal tissue. The communications among these tissues are important for the seed development, which regulatory mechanisms remain unclear. This manuscript entitled "Seed coat-derived brassinosteroids non-cell 3 autonomously regulate endosperm development" revealed that BR regulates cell wall-related processes in the seed coat, and that the biophysical properties of this maternal organ determine the proliferation rate of the endosperm in a manner independent from the timing of its cellularization. And the authors proposed that BR signaling in maternal tissues tunes endosperm proliferation to seed coat expansion. The topic is interesting, and the manuscript shows lots of work. However, there isn't a main thread for the readers to follow, and the logicality is not clear enough. It is difficult to understand the data and conclusion. The organization of this

manuscript needs to be significantly improved. And some conclusions should be turned down to avoid overstatement. Taken together, this manuscript could not be considered for publishing in the Nature Communications unless substantial modification is made, and all the questions raised by reviewers are addressed. The main concerns (not all) are listed below.

Thank you for evaluating our manuscript and for the constructive feedback. We made changes throughout the manuscript to address your concerns and to improve the general readability.

Major points:

1. The authors used fertilization-independent (autonomous) endosperm to study the communications between endosperm and integuments, which would be artificial and could not reflect the real conditions. According to that, the conclusion should be based on this specific condition but not native condition.

Yes, you are correct that we should be more careful with the interpretation of this data. The reason why we tested autonomous seeds, in addition to fertilized ones, was to verify if the induction of endosperm formation by auxin was downstream of BR function (as we demonstrate down the line in the manuscript). However, we agree with you that the conclusions drawn from these experiments are limited, given their artificial nature. These assays also caused confusion to Reviewer 1. Because of this, and because these results did not add much to our conclusions, we decided to remove most data related to auxin-induced autonomous seeds from the manuscript. We only left data which is necessary to understand the design of the RNAseq experiment.

Now, we only focus on sexually produced endosperms, as you recommend.

2. The treatment of BR and auxin in BR-deficient mutant should be very careful. Since BR treatment could complement the phenotype of BR deficiency, there would not be significant difference between the BR and auxin treatment in *det2* and auxin treatment in WT, unless the special design of treatment way. Besides, BR and auxin crosstalk a lot in different levels. It is necessary to exclude the indirect BR influences in auxin regulation of embryo/endosperm/integument development.

Indeed the BR+auxin treatments on *det2* do not fully rescue the phenotype to the level of a WT treated with auxin. Because we are applying the hormones on the pistils, we cannot control how much hormone actually goes into the seeds. Moreover, we only applied the hormones once, and maybe a continuous application would be needed for full restoration of the phenotype.

As you point out, auxin and BR have been shown to interact. And indeed, we do propose that BR signalling does determine the level of auxin activity in the endosperm (as seen from our R2D2 data). However, loss of BR does not seem to affect auxin activity in the seed coat (again based on the R2D2 data). Regarding the embryo, it was previously shown that the endosperm can develop normally without the presence of an embryo

(<https://doi.org/10.1093/plcell/koab007>). Therefore, we do not expect effects of the embryo on our assays.

3. The authors detected the expressions of many components of BR signaling and concluded that BR related genes didn't express during endosperm development. But in figure s1, BZR1 was expressed in embryo. In addition, the data from Arabidopsis eFP browser showed that DET2 and BZR1 were expressed in the embryo. Besides, the authors did not detect BZR1 expression in the central cell or in the endosperm. However, the data from Arabidopsis eFP browser showed that BZR1 was expressed in the endosperm. Based on the unpublished data from the collaborators and I, at least BZR1 clearly expressed in embryo sac including central cell (might not in nuclei). Probably BR is existing and playing roles in endosperm, and BZR1 regulated genes might still work during endosperm development. At least, the possibility could not be excluded. I will show the pictures if necessary. In Summary, the authors need to provide additional data to determine whether BZR1 was expressed in the endosperm or not. Otherwise, the conclusion that "BR regulate endosperm development via the maternal sporophytic tissues" is overstate.

The discrepancy between our reporter data and the eFP browser may have to do with the developmental stage. Our data is based on earlier developmental time-points, as the eFP browser data is based on the Belmonte et al (2013) PNAS manuscript, and the earliest time-point assayed there is at the early globular stage. This is later than the stages we study in this manuscript.

We find it interesting that the reviewer detected BZR1 expression in the central cell. We checked several published BZR1 reporters and never observed it expressed in the gametophyte. While we do agree that reporter data can be unreliable (for example depending on the length of the promoter used in the construct), we still think our interpretation is correct. From the reciprocal crosses of Fig. 3, it is clear that losing BR11 signaling in the sporophyte is sufficient to cause endosperm defects. However, we agree with the reviewer that we cannot fully rule out that BZR1 plays roles in endosperm development, in pathways independent of BR11. We have rephrased our conclusions to reflect that it is BR signaling via BR11 (specifically) that controls endosperm proliferation in a sporophytic manner (lines 147, 172, 263). Therefore, we toned down our conclusions, as you recommended.

We now also add new data on the expression of the BR transporters PGP1 and PGP19. Both ABCB-type transporters were recently shown to transport BR. Coinciding with this, we only see the transporters expressed in the seed coat, and never in the endosperm (see Fig. S3).

4. One of important conclusions is "BR regulate cell wall-related processes in the seed coat, and that the biophysical properties of this maternal organ determine the proliferation rate of the endosperm in a manner independent from the timing of its cellularization", which based on the endosperm cellulation time of *det2-1* and *bri1-6*. Basically, this conclusion would be right, but needs to be limited. The research of endosperm cellulation proposed that the later endosperm cellulation, the larger seeds. The authors cladded that the BR deficient mutants have smaller seeds but no significant different endosperm cellularization time (the time

relative to embryo development, but not the day after fertilization since *det2-1* develops generally slowly than WT), which is consistent to the previous publications (Jiang et al., 2013). However, BR regulates the expression of SHB1, MINI3 and IKU2, and the genetic experiments illustrated these genes work downstream of BZR1 (Jiang et al., 2013). Although it is unclear that why BR regulates the expression of genes related to endosperm cellularization and seed size but did not affect the endosperm cellularization time, the reason that authors mentioned for not supporting the direct BR regulation of endosperm development (BZR1 didn't express in central cell or endosperm) might not be solid enough (see major point 2). Furthermore, endosperm cellularization is still an important regulatory mechanism of seed size. I suggested the authors to carefully discuss this part and turn down the conclusion to something like "our results revealed that BR regulates the proliferation rate of the endosperm through maternal way (cell wall-related processes in the seed coat), and we could not exclude that other ways..."

The reviewer is correct in pointing out that previous research has shown a correlation between endosperm cellularization and seed size. However, that was linked to zygotic effects, for example of MINI3 and IKU2. Sporophytic effects like the ones that we describe here, however, work the other way around: smaller seed coats correlating with early cellularization, and larger seed coats with late cellularization:

<https://link.springer.com/article/10.1007/s00497-009-0116-1>

<https://academic.oup.com/plcell/article/17/1/52/6113136>

Moreover, seed growth arrest has actually been shown to initiate 2 days before the onset of cellularization, meaning that these two parameters can be unlinked, as we propose here:

<https://www.nature.com/articles/s41467-022-35542-5>

The point that we wanted to make here is that sporophytic effects do not always translate into a difference in the timing of cellularization. As you recommend, we now better discuss this in lines 489-505.

5. In the result 1, the *det2-1* and double mutant *br6ox1/2* seeds produce fewer endosperm nuclei compared to WT. A reduction in the rate of autonomous seed formation was observed in the *det2-1* mutant, but not observed in the double mutant *br6ox1/2*. Since DET2 and BR6OX1/2 are key enzymes of BR synthetic pathway, the authors should explain what mechanism led to this difference.

This is a good point, which we also thought was unexpected. In theory, plants lacking BR6ox1 and BR6ox2 should be devoid of brassinolide. However, although the *br6ox1/2* double mutant that we used is a full KO, the plants are viable, although dwarf. This is in contrast with KO mutants of CPD, DWF4 and other BR biosynthesis mutants. In those cases, the plants are barely viable. See Fig. 1b here:

<https://www.nature.com/articles/s41477-021-00917-x>

For instance the *det2* and *dwf4* mutants that we used in this study are hypomorphic mutants, because full KOs are not fertile. To us, this likely signified that there are intermediates in the BR biosynthesis grid that also have important biological roles. Those intermediates may be lacking in strong *det2* or *dwf4* mutants, but not in *br6ox1/2*.

Alternatively, there may be enzymes with redundant functions to BR6ox1/2 that have not been discovered.

In essence, it is not known by the BR field why br6ox1/2 mutants are weaker than other mutants in the BR biosynthesis pathway. As mentioned in the response to point 1, we decided to remove the auxin-induced autonomous seed data from the manuscript, as it was causing confusion and was not adding relevant information to our conclusions.

6. For the endosperm nuclei phenotype, det2 (BR deficient mutant) and bes1-D/bzr1-D (BR signal enhanced mutant) led to similar phenotype. The authors clade that the decrease in nuclei number observed in bzr1-D is likely caused by excessive signaling rather than low endogenous BR. However, they saw no differences between bzr1-D/-/dwf4-5D/- and the single mutant bzr1-D/-, indicating there isn't feedback inhibition of BR synthesis. Thus, how can the authors explain the reason that BR deficient mutant and BR signal enhanced mutant led to similar phenotype? It seemed that BR probably did not play direct roles in this process? These results make the readers confused because they are different from many BR-regulated biological functions published previously. The authors should provide additional data to demonstrate this point.

We argue for a dose-dependent role of BR in regulating endosperm development, where both too high and too low doses of BR are detrimental. This fits with previous observations in roots, in pollen tubes and in the seed coat, where different cell types have different demands for BR concentrations and too high concentrations are deleterious for growth:

<https://www.nature.com/articles/s41477-021-00917-x>

<https://link.springer.com/article/10.1007/s00497-014-0247-x>

<https://www.biorxiv.org/content/10.1101/2023.12.07.569203v3>

We mention this in the introduction in lines 57-70 and discuss it in lines 469-473.

It is not known in the literature why too much BR can both have deleterious effects on development. It is similar to what is seen for auxin, where too high concentrations can be inhibitory for growth, but the molecular mechanisms remain undiscovered.

Minor points:

1. The descriptions like “We hypothesized that... This was based on...” and “We then hypothesized that... Therefore, we analyzed...” didn't match the normal reading habits of research article and has strong potential assumption.

While we do note the opinion of the reviewer, we do not feel that the way we present our hypotheses is wrong. We generated and reformulated hypotheses, depending on the outcomes of given experiments, and then proceeded to test them. We do not see a problem with our wording, but are happy to revise specific statements, in case they are ambiguous.

2. There are too many clauses in this manuscript, which made sentences too long and difficult to understand.

We simplified some of the sentences in the manuscript.

3. The figures are a little bit preliminary. It is much helpful to label important parts in the pictures, such as the endosperm nucleus in Fig1b and c.

Agreed, we added this information to the figures.

4. The figure legends are too simple. The detailed information is needed to understand the results, such as the method of statistical analysis.

We feel that the legends are already quite long, and therefore we only provide the necessary information for the results to be interpreted. Experimental details, including full explanation of the statistical analyses, are found in the Materials and Methods.

Reviewer #3 (Remarks to the Author):

This manuscript by Lima et al., reports that seed coat-derived brassinosteroids non-cell autonomously regulate endosperm proliferation. They suggest that BR does so by affecting auxin signaling and by regulating cell-wall softening. This is an interesting finding; however, the current data are not sufficient to support the main conclusion. My comments are listed below:

Thank you for the constructive comments. We modified the manuscript to address your concerns.

1. If BR is required for endosperm proliferation, the mutations resulting constitutive BR signaling should lead to increased endosperm proliferation. However, *bes1-D* and *bzr1-D* actually showed a reduction of endosperm proliferation. These observations do not support the author's conclusion and should be carefully interpreted.

We actually propose that BRs regulate endosperm proliferation in a dose-dependent manner, where too low or too high concentrations are detrimental. This fits with previous observations in roots, in pollen tubes and in seed coats, where different cell types have different demands for BR concentrations and too high concentrations are deleterious for growth:

<https://www.nature.com/articles/s41477-021-00917-x>

<https://link.springer.com/article/10.1007/s00497-014-0247-x>

<https://www.biorxiv.org/content/10.1101/2023.12.07.569203v3>

We mention this in the introduction in lines 57-70 and discuss it in lines 469-473.

2. Genes involved in BR perception, biosynthesis and TFs are expressed in the endosperm (Fig S1). Thus, BR biosynthesis in the endosperm might impact endosperm development. So the observed reduction of endosperm proliferation in BR mutants might not because of lack of cell coat-derived BR.

The data we show in Fig S1 is based on datasets by Belmonte *et al.* (2013). These datasets are based on microarray experiments done from laser microdissected materials. However, the time points assessed in those datasets are from later stages of seed development, compared to the ones we study here. Therefore, we cannot rule out that some of those genes are expressed in zygotic products at later stages than those assessed in this manuscript. Moreover, because this

dataset is based on microarray data, it is not 100% reliable. For instance, genes that are known to be specifically expressed in seed coats, like BAN, TTG2, STK, CHS, among many others, are found as “expressed” in the embryo and endosperm samples of the Belmonte (2013) dataset. This likely indicates contamination of embryo and endosperm samples by sporophytic tissues, or may also be an artifact of the microarray, which relies on fluorescence and not on number of reads, like RNAseq does. Therefore, although these datasets are still a useful primary source for identifying genes expressed in seed compartments, validation using reporter lines and genetic experiments, always has to complement it. This is exactly what we did, and we believe we present strong enough genetic evidence to conclude that BRs originating in the sporophyte are influencing endosperm proliferation.

3. BR mutants have reduced auxin activity in the endosperms, however, no evidence was provided to show that it is the reduced auxin activity that leads to decreased endosperm proliferation.

This is based on earlier findings. We previously showed that reduced auxin activity in endosperms leads to reduced proliferation:

<https://www.nature.com/articles/nplants2015184>

We mention this in line 341, and make it more clear in lines 539-541.

4. BR mutants have stiffer seed coats. However, no data was provided to support that stiffer seed coats contribute to the reduced endosperm proliferation. Cell wall softening mutants might be used to cross with BR mutants and then check the endosperm proliferation phenotype. Several unrelated mutants that produce smaller seed coats have reduced proliferation; however, because they are selected; it is still possible that other mutants with smaller seed coats might have normal endosperm proliferation.

Regarding the stiffness of the seed coat, we believe we address this question with the EGCG treatments of Fig. S8. EGCG “relaxes” the cell walls, and this leads to a rescue of the BR mutant phenotypes. We also added the analysis of the *xx1/2* mutant, which lacks xyloglucans, and whose cell walls have therefore different properties from the WT ones.

Regarding the mutants that we tested, they were not “selected”. We obtained mutants that were described to have smaller seed coats. We have so far not encountered any mutant producing smaller seed coats at 2 DAP, which does not also show reduced endosperm proliferation. This fits perfectly with our hypothesis.

Moreover, we actually obtained additional mutants that were reported to produce smaller seed coats, like *stk* and *ttg2*, but we could not see any size differences in seed size at 2 DAP.

Therefore we could not include them in this analysis.

Reviewer #1 (Remarks to the Author):

The authors have addressed all my concerns.

Thank you for the constructive criticism. Your comments very much improved our manuscript.

Reviewer #2 (Remarks to the Author):

The revised version has been improved in logicality and organization. Some points became clearly.
Thank you.

However, they remove most data related to auxin induced autonomous seeds from the manuscript, leading to reduced depth of mechanism and innovation.

We agree. But as you pointed out in the previous version of the manuscript, that data was not easy to interpret and to conciliate with the data originating from sexual seeds. We however left enough data showing that BRs do affect asexual seed formation.

Furthermore, the author wrote the hypothesis first, then described the results later, which gives readers a impression that the author presumed the conclusions and subsequently demonstrated them. In my opinion, it would be better to modify the descriptions.

We do not follow this logic. This is how the scientific method works. We generate a hypothesis and then proceed to test it. We in fact disproved some of our initial hypotheses throughout the manuscript. We do not see what the alternative structure would be.

Reviewer #3 (Remarks to the Author):

The authors made some changes in response to my previous concerns. However, the conclusion "BR regulate endosperm development via the maternal sporophytic tissues" and the model in figure 6f remain seriously misleading. The authors have not adequately address whether BR exists and plays a role in tissues other than the seed coat, such as endosperm. Data in Figure 3d and 3f showed that reconstruction of BR signaling in the seed coat only marginally rescued of the det2-1 and bri1-6 endosperm phenotypes, indicating that BR signaling in tissues other than the seed coat is playing even more essential roles in endosperm development than seed coat does. Consistent with this notion, the gene expression data showed that BR pathway genes are expressed in endosperm, as also raised by reviewer 2. The title should be revised to avoid implying that BR affects endosperm development exclusively via the seed coat.

We agree that our conclusions were sometimes too strong. We have now modified the text throughout the manuscript, including the title, to not exclude the possibility that BRs originating in other tissues may also contribute to endosperm formation.

The authors suggested a dose-dependent mechanism to explain why constitutive BR signaling caused unexpected reduction of endosperm proliferation. However, this explanation contradicts findings from the BRI1 overexpression line, which did not show reduced endosperm proliferation. These conflicting results do not support the authors' conclusions.

We believe this has to do with how strongly the BR signalling is activated. The BRI1ox line is simply an expression of BRI1 using its endogenous promoter in the WT background. This is used in the literature as a BR signalling overproducer. But in fact the phenotype of these plants is weaker than

the phenotypes of bzc1-d or bes1-d lines. Those two have much stronger developmental defects. We think this is the reason why we only detect strong phenotypes in those two lines, but not in BRI1ox.

In sum, I recommend the authors be careful when interpreting their data and to tone down their conclusions throughout the manuscript.

Thank you. We did as you suggest.